

# Drivers of decadal trends of the ocean carbon sink in the past, present, and future in Earth system models

Jens Terhaar[1,2,3]

[1]Woods Hole Oceanographic Institution, Woods Hole, MA, USA
5  [2]Climate and Environmental Physics, Physics Institute, University of Bern, Bern, Switzerland
[3]Oeschger Centre for Climate Change Research, University of Bern, Bern, Switzerland

*Correspondence to*: Jens Terhaar (jens.terhaar@unibe.ch)

**Abstract.** The land biosphere and the ocean are the two major sinks of anthropogenic carbon at present. When anthropogenic carbon emissions become zero and temperatures stabilizes, the ocean is projected to become the dominant and only global natural sink of carbon. Despite the ocean's importance for the carbon cycle and hence the climate, observing the ocean carbon sink and detecting anthropogenic changes over time remain challenging because uncertainties of the decadal variability of this carbon sink and the underlying drivers of this decadal variability remain large. The main tools that are used to provide annually resolved estimates of the ocean carbon sink over the last decades are global observation-based $pCO_2$ products that extrapolate sparse $pCO_2$ observations in space and time and global ocean biogeochemical models forced with atmospheric reanalysis data. However, these tools (i) are limited in time over the last 3 to 7 decades, which hinders statistical analyses of the drivers of decadal trends, (ii) are all based on the same internal climate state, which makes it impossible to separate externally and internally forced contributions to decadal trends, and (iii) cannot assess the robustness of the drivers in the future, especially when carbon emissions decline or cease entirely. Here, I use an ensemble of 12 Earth System Models (ESMs) from phase 6 of the Coupled Model Intercomparison Project (CMIP6) to understand drivers of decadal trends of the past, present and future ocean carbon sink. The simulations by these ESMs span the period from 1850 to 2100 and include 4 different future Shared Socioeconomic Pathways (SSPs), from low emissions and high mitigation to high emissions and low mitigation. Using this ensemble, I show that 80% of decadal trends in the multi-model mean ocean carbon sink can be explained by changes in decadal trends of atmospheric $CO_2$ as long as the ocean carbon sink remains smaller than 4.5 Pg C yr$^{-1}$. The remaining 20% are due to internal climate variability and ocean heat uptake, which results in a loss of carbon from the ocean. When the carbon sink exceeds 4.5 Pg C yr$^{-1}$, which only occurs in the high emission SSP3-7.0 and SSP5-8.5, atmospheric $CO_2$ rises faster, climate change accelerates, the ocean overturning and the chemical capacity to take up carbon from the atmosphere reduce, so that decadal trends in the ocean carbon sink become substantially smaller than estimated based on changes in atmospheric $CO_2$ trends. The breakdown of this relationship in both high emission pathways also implies that the decadal increase in the ocean carbon sink is effectively limited to be ~1 Pg C yr$^{-1}$ dec$^{-1}$ in these pathways, even if the trend in atmospheric $CO_2$ continues to increase. Previously proposed drivers, such as the atmospheric $CO_2$ or the growth rate of atmospheric $CO_2$ can explain trends in the ocean carbon sink for specific time periods, for example



during exponential atmospheric $CO_2$ growth, but fail when emissions start to decrease again. The robust relationship over a large ESM ensemble also suggests that very large positive and negative decadal trends of the ocean carbon sink by some $pCO_2$ products are highly unlikely, and that the change in the decadal trends of the ocean carbon sink around 2000 is likely

substantially smaller than estimated by these $pCO_2$ products.

**1 Introduction**

The ocean has taken up around one quarter of all anthropogenic $CO_2$ emissions from land use change and fossil fuels since the beginning of the industrial revolution (Friedlingstein et al., 2023; Gruber et al., 2023; Terhaar et al., 2022b). As such, it

is, in addition to the land biosphere, one of the two major natural sinks of carbon in the earth system. Once temperatures stabilize, the ocean will become the dominant global natural sink of carbon (Silvy et al., 2024) and will store more than half of the anthropogenically emitted $CO_2$ in around 1000 years (Joos et al., 2013). By taking up carbon from the atmosphere, the ocean effectively slows down global warming (IPCC, 2021) and will contribute to stabilizing global temperatures over the next centuries if emissions reach near-zero (Terhaar et al., 2023; MacDougall et al., 2020). Here we define the ocean carbon

sink as in the Global Carbon Budget as the change in air-sea $CO_2$ flux due to anthropogenic carbon emissions and anthropogenic climate change in comparison to a relatively stable pre-industrial state (Friedlingstein et al., 2023). Consequently, 'anthropogenic' refers to direct effects from anthropogenic emissions and the indirect effect to the anthropogenically caused climate change.

The overall magnitude of the ocean carbon sink is mainly determined by the ocean overturning circulation, i.e., the rate at which surface waters with increased anthropogenic carbon content can be transported to the deep ocean and be replaced by waters with low anthropogenic carbon content (Sarmiento et al., 1992; Caldeira and Duffy, 2000; Orr et al., 2001). Furthermore, the magnitude of the ocean carbon sink is influenced by surface ocean capacity to take up more anthropogenic carbon, which itself is determined by the surface ocean carbonate chemistry and especially the alkalinity (Broecker et al.,

1979; Terhaar et al., 2022b). Over the historical period, the change in atmospheric $CO_2$ has been the main driver of changes in the ocean carbon sink and is assumed to be approximately proportional to the strength of the ocean carbon sink (Mikaloff Fletcher et al., 2006; Gruber et al., 2009; Terhaar et al., 2021b). However, this linear relationship between the strength of the ocean carbon sink and atmospheric $CO_2$ is only assumed to work under exponential atmospheric $CO_2$ growth (Raupach, 2013; Raupach et al., 2014).


Over the last decades, the relatively steady growth of the ocean carbon sink has been weakened by outgassing of natural carbon due to warming and climate change (Joos et al., 1999; McNeil and Matear, 2013; Frölicher et al., 2015) and is superimposed by decadal variability and trends of the ocean carbon sink, i.e., a reduction in the 1990s and an increase since



2000 (Lovenduski et al., 2008, 2007; Le Quéré et al., 2007; Landschützer et al., 2015, 2016). A consensus of the drivers of
these trends is still not reached and possible explanations for these different trends are changes in wind and pressure systems
(Le Quéré et al., 2007; Keppler and Landschützer, 2019), variability in the ocean circulation and ventilation (DeVries et al.,
2017), or changes in the atmospheric $CO_2$ growth rate and ocean surface temperature due to the eruption of Mount Pinatubo
(McKinley et al., 2020; Frölicher et al., 2011). Moreover, recent studies suggest that the observational-based decadal
variability over the last decades might be overestimated (Gloege et al., 2021; Hauck et al., 2023).


Despite the importance of the ocean carbon sink for the global climate and carbon cycle, observing or simulating the ocean
carbon sink is still challenging. The two main tools to estimate the annually-resolved ocean carbon sink over the past four to
seven decades, to provide an annual update every year within the Global Carbon Budget, and to understand the drivers of the
magnitude and trends of the ocean carbon sink are observation-based $p$CO$_2$ products and global ocean biogeochemical
models (GOBMs) (Friedlingstein et al., 2023; Hauck et al., 2023). Observation-based $p$CO$_2$ products extrapolate relatively
sparse observations of surface ocean $p$CO$_2$ using statistical methods and/or machine learning to create global monthly maps
of surface ocean $p$CO$_2$ (Fay et al., 2021; Gregor and Gruber, 2021; Chau et al., 2022; Rödenbeck et al., 2015; Watson et al.,
2020; Landschützer et al., 2015). These monthly $p$CO$_2$ maps are then used to estimate the global ocean carbon uptake. In
addition, an estimate of pre-industrial natural outgassing of $CO_2$ due to the difference of riverine carbon input and carbon
sequestration in sediments (Regnier et al., 2022) has to be added to estimate the change in the air-sea $CO_2$ flux with respect
to pre-industrial conditions. Global ocean biogeochemical models (GOBMs) (Orr et al., 2001; Hauck et al., 2020; Terhaar et
al., in press) simulate the ocean carbon sink while being forced with past observed atmospheric $CO_2$ and observation-based
reanalysis data, such as wind, humidity, precipitation, temperatures (Hersbach et al., 2020; Tsujino et al., 2018).

The estimates of both product classes vary in magnitude and decadal trends, with $p$CO$_2$ products estimating a larger
magnitude of the ocean carbon sink and also generally larger decadal trends over the last two decades (DeVries et al., 2023;
Friedlingstein et al., 2023). One reason for the low carbon sink in $p$CO$_2$ products might be the starting year of these
simulations that is often later than the beginning of the simulation and the thus slightly too high different pre-industrial
reference period and $p$CO$_2$ in the ocean (Terhaar et al., in press; Bronselaer et al., 2017). Another reason is existing biases in
the simulated ocean circulation, especially the Southern Ocean and Atlantic Ocean overturning, and biases in the surface
ocean carbonate chemistry in GOBMs that were previously identified as drivers of uncertainties and biases in ESM
ensembles (Terhaar et al., 2022b, 2021b; Goris et al., 2018). As opposed to the magnitude, the differences in the decadal
trends between both products might be due to sparse amount of observations in space and in time, especially in the 1980s
and 1990s, as demonstrated with a subset of $p$CO$_2$ products evaluated with output from a GOBM (Hauck et al., 2023; Gloege
et al., 2021). In addition to differences in trends between from $p$CO$_2$ products and GOBMs, no consensus has yet been made
with respect to the underlying drivers of the decadal trends in the ocean carbon sink (Friedlingstein et al., 2023; DeVries et
al., 2023; Gruber et al., 2023). The detection of these drivers with $p$CO$_2$ products, GOBMs, and other methods such as data



assimilation models (DeVries et al., 2017), is difficult due to the relatively short time period over which enough $p\text{CO}_2$ observations and atmospheric reanalysis data exist, due to the relative homogeneity of drivers over this period, e.g., constantly increasing atmospheric $\text{CO}_2$, and the absence of an alternative climate state against which these ocean carbon sink estimates can be compared.

Here, I use an ensemble of 12 ESMs to provide a new perspective on potential drivers of the decadal trends of the ocean carbon sink from phase 6 of the Coupled Model Intercomparison Project (CMIP6) (Table 1). For the analyses of decadal drivers of the ocean carbon sink, ESMs have distinctive advantages compared to $p\text{CO}_2$ products and GOBMs because (1) they cover a period of 251 years from 1850 to 2100, (2) cover at least four different future scenarios, and (3) they all have a different internal climate state. The long time-period with different climate states in each model gives ample material to perform statistical analyses and the different future scenarios allow to test the robustness of drivers of the decadal variability of the ocean carbon sink under continuously rising and under strongly decreasing carbon emission trajectories. Furthermore, using different Earth System Models in comparison to large ensembles of one ESM (Fay et al., 2023; McKinley et al., 2016) avoids the risk of having a common bias in that one ESM, which might wrongly influence the analysis. Using the ESM ensemble from CMIP6, I will present how potential drivers of the ocean carbon sink, i.e., the atmospheric $\text{CO}_2$ and its growth rate, ocean heat uptake, and climate variability drive trends in the ocean carbon sink from 1850 to 2100 in these models.



## 2 Methods and Datasets

### 2.1 Earth system model ensemble

In thus study, I use an ensemble of 12 ESMs from CMIP6 (Table 1). All ESMs from CMIP6 that provide the necessary model output for the following analysis were chosen.


**Table 1**: CMIP6 models used in this study and the corresponding model groups.

| Model name | Modelling centre | References |
|---|---|---|
| ACCESS-ESM1-5 | Commonwealth Scientific and Industrial Research Organisation (CSIRO) | (Ziehn et al., 2020) |
| CanESM5 | Canadian Centre for Climate Modelling and Analysis | (Christian et al., 2022) |
| CanESM5-CanOE | | |
| CESM2 | Community Earth System Model contributors | (Danabasoglu et al., 2020) |
| CESM2-WACCM | | |
| CMCC-ESM2 | Centro euro-Mediterraneo sui Cambiamenti Climatici | (Lovato et al., 2022) |
| IPSL-CM6A-LR | Institut Pierre Simon Laplace (IPSL) | (Boucher et al., 2020) |
| MPI-ESM1-2-HR | Max-Planck-Institute for Meteorology | (Mauritsen et al., 2019; Gutjahr et al., 2019) |
| MPI-ESM1-2-LR | | |
| NorESM2-LM | Norwegian Climate Centre | (Tjiputra et al., 2020) |
| NorESM2-MM | | |
| UKESM1-0-LL | Met Office Hadley Centre | (Sellar et al., 2020) |

### 2.2 Calculating the ocean carbon sink

The annually averaged ocean carbon sink was calculated from concentration-driven historical simulations from CMIP (1850-
2014) and four different concentration-driven Shared Socioeconomic Pathways (SSPs) (2015-2100): the low-emission high-mitigation SSP1-2.6, the high-emission low-mitigation SSP5-8.5, and the two intermediate pathways SSP2-4.5 and SSP3-7.0 (Riahi et al., 2017). To account for drifts in the historical and SSP simulations, a linear fit was calculated over the annual carbon sink over the years of the pre-industrial control run that correspond to the years 1850 to 2100 in the historical and SSP simulations. The linear change in the carbon sink in the pre-industrial simulations since 1850 was then subtracted from
the historical and SSP simulations.

Furthermore, ESMs have biases in the magnitude of the ocean carbon sink due to biases their respective circulation and surface ocean carbonate chemistry that also affect the size of the decadal trends, i.e., a negative bias in the magnitude of the carbon sink also introduces a negative bias in the decadal trends. To statistically compare the decadal trends of the carbon
sink over the here-used ESM ensemble, the global estimate of the ocean carbon sink was adjusted for each ESM with respect





to biases in its circulation and surface ocean carbonate chemistry following Terhaar et al. (2022). First, the Revelle factor, the inter-frontal Southern Ocean sea surface salinity, and the AMOC strength were calculated for each model. Afterwards, a multi-linear fit was performed with the three observation-based quantities as predictors and the average ocean carbon sink from 1850 to 2100 as target variables (the period from 2015 to 2100 was used four times for each of the four SSPs). Finally,

the biases in each predictor with respect to observation-based estimates of these predictors are calculated and used to adjust the simulated ocean carbon sink based on the determined constants from the multi-linear fit. Overall, this result in an adjustment of $10 \pm 7\%$ (i.e., increased ocean carbon uptake) for the here used model ensemble.

The ocean carbon sink was also calculated for each of the five major ocean basins (Atlantic Ocean, Pacific Ocean, Indian

Ocean, Arctic Ocean, and Southern Ocean) using the RECCAP2 biome mask (DeVries et al., 2023; Terhaar et al., in press), which is a slightly adapted version of a previously developed biome mask (Fay and McKinley, 2014). Regionally, no bias adjustments were performed as it still remains largely unclear how biases in circulation and carbonate chemistry affect the regional ocean carbon sink estimates.

### 2.3 Atmospheric CO$_2$ and growth rate

The annually averaged atmospheric CO$_2$ over the historical period and for each SSP was taken from the CMIP6 forcing files (Meinshausen et al., 2020, 2017). The atmospheric CO$_2$ growth rate in each year was calculated as the difference in atmospheric CO$_2$ in that year and the year before.

### 2.4 Estimating the effect of climate change and ocean heat uptake on the ocean carbon sink

The effect of climate change and ocean heat uptake on the ocean carbon sink in ESMs was calculated based on additional

idealized simulations provided by five of the twelve ESMs in the ensemble (ACCESS-ESM1-5, CanESM5, MRI-ESM2-0, NorESM2-LM, UKESM1-0-LL) within the CMIP6 framework. These five ESMs made historical simulations, called 'bgc', where the change in atmospheric CO$_2$ had no effect on climate change but the carbon cycle still 'sees' the increase in atmospheric CO$_2$. However, other non-CO$_2$ radiative agents (aerosols, CH$_4$, N$_2$O, etc.) still effect the climate in these simulations. These 'bgc' simulations were only made for SSP5-8.5 ('ssp585-bgc') and not for the other SSPs. The difference

of the normal historical and SSP5-8.5 simulations (including effects from CO$_2$ and non-CO$_2$ radiative agents) and the additional 'bgc' simulations quantifies the heat and carbon fluxes that are solely due to the CO$_2$-induced climate change and warming.

### 2.5 Climate modes

To assess the climate variability across the ensemble of the ESM, annual averages of three climate modes were calculated for each ESM over the 251 years of the pre-industrial control simulation: (1) The Atlantic Multi-decadal Oscillation (AMO), (2)





the Niño 3.4 index, and (3) the Marshall Southern Annular Mode (SAM) index. The AMO was calculated based on SST anomalies in the North Atlantic between 0 and 80°N. The Niño 3.4 index was calculated based on SST anomalies in the tropical Pacific region from 5°S to 5°N and from 170°W to 120°W. The Marshall SAM index was calculated as anomalies of

the zonal pressure difference between the latitudes of 40S and 65S. Anomalies for each index in ESMs were calculated by removing a linear fit over the 251 years of the pre-industrial control simulation.

In addition, observation-based estimates of each climate mode were used. The observation-based AMO index (https://climatedataguide.ucar.edu/sites/default/files/2022-03/amo_monthly.txt) and the Niño 3.4 index

(https://psl.noaa.gov/gcos_wgsp/Timeseries/Data/nino34.long.data) are based on HadISST1 (Rayner et al., 2003). The Marshall SAM index has been calculated based on twelve stations, six stations at ~40°S and six stations at ~65°S (Marshall, 2003). To compare each observation-based index to the simulated index in the pre-industrial control simulations, the timeseries of the observation-based indexes have been detrended by subtracting a linear trend over the respective observation-based index estimate.

**2.6 Decadal trends**

Decadal trends of different variables are here defined as the slope of linear fits over ten years.



## 3 The influence of atmospheric $CO_2$ on the ocean carbon sink

### 3.1 Atmospheric $CO_2$

Over the historical period of CMIP6 simulations from 1850 to 2014, the annually averaged global ocean carbon sink has increased approximately proportional to the rise in atmospheric $CO_2$ (Figure 1a, b, c). Due to the exponential rise in atmospheric $CO_2$, the cumulative ocean carbon sink is also approximately proportional to the rise in atmospheric $CO_2$. However, these quasi-linear relationships did not hold from 1920 (atmospheric $CO_2$ of 304 ppm) to 1960 (317 ppm) and from 1990 (354 ppm) to 2000 (369 ppm) when the ocean carbon sink did not increase while the atmospheric $CO_2$ continued

to increase. These periods manifest themselves as 'jumps' in the linear relationship between the atmospheric $CO_2$ and the cumulative ocean carbon sink (Figure 1d).

After 2014, the link between atmospheric $CO_2$ and the ocean carbon sink depends strongly on the future scenario of atmospheric $CO_2$. The linear relationship between the annually averaged carbon sink and atmospheric $CO_2$ breaks down

under all scenarios. Under SSP5-8.5, a pathway with continuous increase in emissions (Riahi et al., 2017) and exponentially growing atmospheric $CO_2$ (Fig. 1a), the increase in the ocean carbon sink per increase in atmospheric $CO_2$ reduces until the ocean carbon sink reaches a maximum just above 6 Pg C yr$^{-1}$ (Fig 1b), which is not exceeded even if atmospheric $CO_2$ rises (Fig 1e). Under SSP3-7.0, a pathway with slightly smaller emissions and atmospheric $CO_2$ than SSP5-8.5, the ocean carbon sink also converges to a maximum but at around 5 Pg C yr$^{-1}$. Under SSP2-4.5, $CO_2$ emissions start to decline around 2050

(Riahi et al., 2017) and atmospheric $CO_2$ stabilizes around 600 ppm by 2100 (Fig 1a). Although atmospheric $CO_2$ stabilizes, the ocean carbon sink reduces strongly (Fig 1b). Under SSP1-2.6, atmospheric $CO_2$ does not only stabilize but starts to reduce by 2080, leading to a strong reduction of the ocean carbon sink (Fig. 1a). In comparison, the relationship between the cumulative ocean carbon sink and atmospheric $CO_2$ remains almost linear in the two high-emission pathways (SSP3-7.0 and SSP5-8.5) although the slope reduces with warming (Fig. 1f). For the two low-emission pathways (SSP1-2.6 and SSP2-4.5),

the relationship breaks down as the ocean continuously takes up carbon, even when atmospheric $CO_2$ stabilizes and decreases (Fig. 1e, f).





**Figure 1: The relationship between atmospheric $CO_2$ and the global ocean carbon sink. (a)** The annually averaged atmospheric $CO_2$ that was used to force the ESMs from CMIP6 based on observation-based estimates from 1850 to 2014 (black) and based on four different SSPs (SSP1-2.6 in blue, SSP2-4.5 in orange, SSP3-7.0 in red, and SSP5-8.5 in brown) from 2015 to 2100. **(b)** The resulting ocean carbon sink as simulated by 12 ESMs (Table 1) after being adjusted for biases in circulation and surface ocean carbonate chemistry following Terhaar et al. (2022). The thick lines indicate multi-model means and the shading the 1-σ standard deviation across the model ensemble. Relationships between atmospheric $CO_2$ and the annually averaged ocean carbon sink **(c)** for the historical period until 2014 and **(e)** for the 21st century from 2015 onwards, as well as between atmospheric $CO_2$ and the cumulative ocean carbon sink **(d)** for the historical period until 2014 and **(f)** for the 21st century from 2015 onwards.



## 3.2 Atmospheric CO₂ growth rate

As an alternative to the atmospheric $CO_2$, the growth rate of atmospheric $CO_2$ was proposed as a key driver for the strength of the ocean carbon sink (McKinley et al., 2017, 2020). Over the historical period, the atmospheric $CO_2$ growth rate appears to be related to the square root of the strength of the ocean carbon sink (Fig 2c). This relationship weakens after ~1960 when
the prescribed atmospheric $CO_2$ growth rate is based on direct atmospheric $CO_2$ observations and not, as before, on relatively smooth observation-based estimates from proxies (Fig. 2a). The direct observations capture the strong inter-annual variability of the atmospheric $CO_2$ growth rate that cannot be reconstructed by observation-based estimates from proxies. However, even this relationship between the atmospheric $CO_2$ growth rate and the square root of the strength of the ocean carbon sink breaks down in the 1920s and 1940s (Fig. 2c, d) when the growth rate is around zero over around a decade each
time (Fig. 2a), but the ocean carbon sink does not go back close to zero but remains almost stable (Fig. 2b).

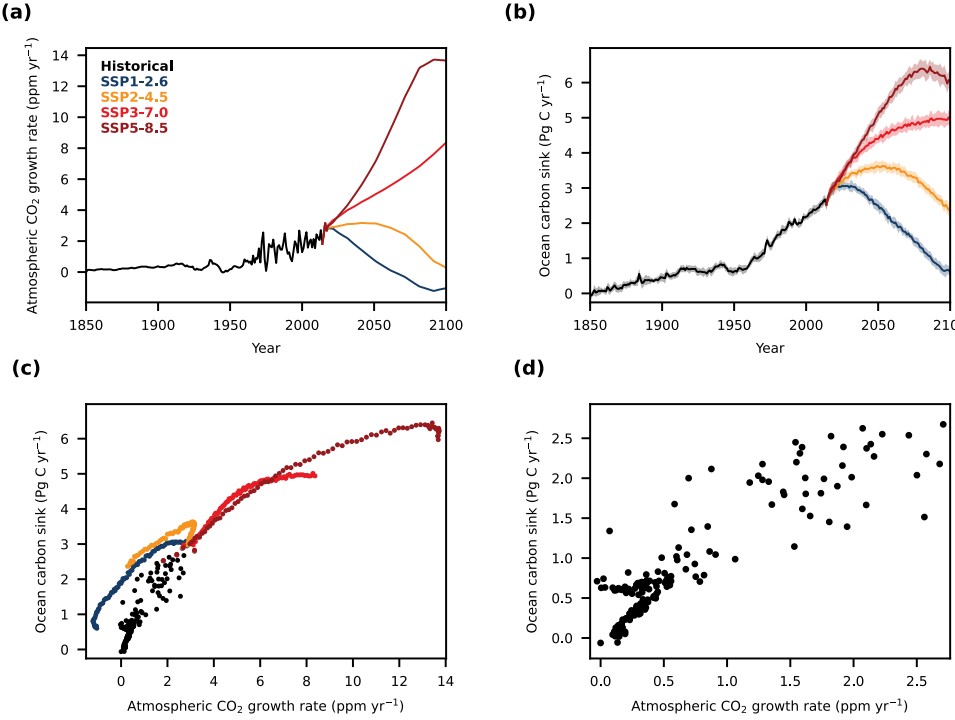

**Figure 2: The relationship between the atmospheric CO₂ growth rate and the global ocean carbon sink. (a)** The annually averaged atmospheric $CO_2$ growth rate based on atmospheric $CO_2$ forcing files from CMIP6, which are based on observation-based estimates from
1850 to 2014 (black) and based on four different SSPs (SSP1-2.6 in blue, SSP2-4.5 in orange, SSP3-7.0 in red, and SSP5-8.5 in brown) from 2015 to 2100. **(b)** The ocean carbon sink as simulated by 12 ESMs (Table 1) after being adjusted for biases in circulation and surface ocean carbonate chemistry following Terhaar et al. (2022). The thick lines indicate multi-model means and the shading the 1-σ standard deviation across the model ensemble. Relationships between atmospheric $CO_2$ growth rate and the annually averaged ocean carbon sink **(c)** for the entire period from 1850 to 2100 and **(d)** only for historical period until 2014.



Over the 21st century, the relationship between the ocean carbon sink and the square root of the atmospheric $CO_2$ growth rate depends on the scenario and breaks down under the two low emission scenarios (SSP1-2.6 and SSP2-4.5) (Fig 2c). As long as $CO_2$ emissions and atmospheric $CO_2$ growth rate rise, as they do under SSP3-7.0 and SSP5-8.5 (Fig 2a), the relationship holds (Fig. 2a). However, the strength of the relationship varies between SSP3-7.0 and SSP5-8.5. Under SSP5-8.5, the relationship also breaks down in the last two decades when the atmospheric $CO_2$ growth stabilizes but the ocean carbon sink
weakens. Under SSP1-2.6 and SSP2-4.5 and declining emissions and declining and even negative atmospheric $CO_2$ growth rates, the ocean carbon sink reduces but not along the same path as it increased over the historical period (Fig. 2c).

### 3.3 Changes in atmospheric $CO_2$ growth rate determine changes in decadal trends of the ocean carbon sink

#### 3.3.1 Global relationship

Although neither the atmospheric $CO_2$ nor its growth rate can quantify the strength of the ocean carbon sink various time period and different trajectories of atmospheric $CO_2$, the atmospheric $CO_2$ growth rate can nevertheless be used to understand changes in the ocean carbon sink on decadal timescales, i.e., decadal trends of the ocean carbon sink. For the period from 1980 to 2018, it has been shown that a smaller increase of the growth rate in comparison to a linear trend has led to a stagnation of the increase of the ocean carbon sink and that an accelerated increase of the growth rate has led to a
strongly increasing carbon sink (McKinley et al., 2020).

Over longer time periods and different future SSPs, ESMs provide more such examples where changes in the growth rate of atmospheric $CO_2$ led to changes in the decadal trends of the simulated ocean carbon sink (Figure 2). Around 1915, the atmospheric $CO_2$ growth rate changes from an increase to a decrease, at the same time the ocean carbon sink stops increasing
and starts to decrease. Then in 1930, the atmospheric $CO_2$ growth rate increases, and the ocean carbon sink also starts to increase simultaneously. Then, in 1940 the atmospheric $CO_2$ growth rate decreases again, and the ocean carbon sink also decreases at the same time. Similarly, the atmospheric $CO_2$ growth rate changes from a positive trend to a negative trend in 1990, exactly when the ocean carbon sink also starts to slow down. When the atmospheric $CO_2$ growth rate increases again, the ocean carbon sink also increases. Over the 21st century, the same relationship continues. Under SSP2-4.5, the
atmospheric $CO_2$ growth rate slows done until 2050 and the positive trend in the ocean carbon sink weakens. Once the atmospheric $CO_2$ growth declines, the trend in the ocean carbon sink becomes negative. However, over longer time periods and under decreasing atmospheric $CO_2$,.

Although a comparison to a theoretical linear trend as in McKinley et al. (2020) is not anymore possible over longer time
periods and under decreasing atmospheric $CO_2$, a clear relationship ($r^2$=0.91) emerges indeed over the entire historical period and all four future scenarios over the 21st century (excluding years where the ocean carbon sink exceeds 4.5 Pg C $yr^{-1}$)



between changes in the atmospheric $CO_2$ growth rate from one decade to another and the decadal trend of the multi-model average of the ocean carbon sink (Fig. 3). Changes in the atmospheric $CO_2$ growth rate are here defined as the change in trends (linear fit over one decade) of atmospheric $CO_2$ from one decade to the next. Trends in the ocean carbon sink are a

linear fit over annual values of the global ocean carbon sink in the 2nd decade. It thus appears that it is the change in the growth rate in comparison to the previous decade that appears to drive the decadal trends of the ocean carbon sink and not the difference to an expected linear trend. If, for example, the growth rate strongly reduced from one decade to another, the ocean carbon sink would show a negative trend. If the growth rate than stays at that lower level, the carbon sink would not decline further but stabilize at its new level. This relationship even holds when $CO_2$ emissions decline strongly as under

SSP1-2.6.

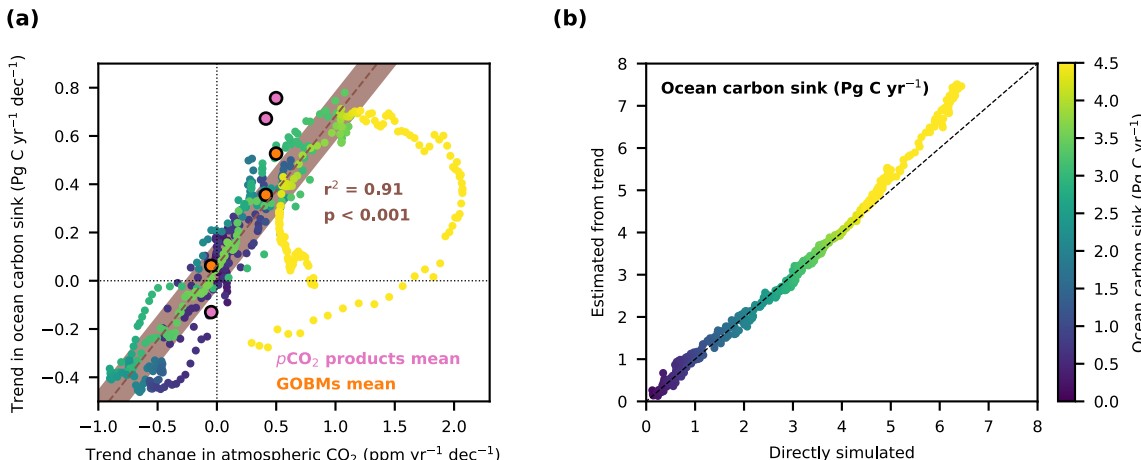

**Figure 3: The relationship between changes in the atmospheric $CO_2$ growth rate and decadal trends of the global ocean carbon sink for the multi-model mean. (a)** Decadal trends of the multi-model mean ocean carbon sink compared to changes in decadal trends in
atmospheric $CO_2$, which represent the decadal averaged growth rate of atmospheric $CO_2$. The dark blue to yellow circles without a surrounding black line show multi-model averages for all years of the historical period from 1850 to 2014 and for all years from 2015 to 2100 for all four SSPs. All decades over from 1850 to 2100 are shown, i.e., 2000-2009, 2001-2010, 2002-2011, etc. The brown line shows a linear fit for all years when the global ocean carbon sink is smaller than 4.5 Pg C yr$^{-1}$ and the brown shading is the 1-σ projection uncertainty. The dots with black lines around them show values from the respective ensemble means of the $p$CO$_2$ products (pink) and
GOBMs (orange) from the Global Carbon Budget 2023 (Friedlingstein et al., 2023) for the three decades between 1990 and 2020. **(b)** The simulated ocean carbon sink in comparison to the expected ocean carbon sink based on the relationship in **(a)** and the prescribed trend change in atmospheric $CO_2$ in the simulations.

However, this relationship between changes in the atmospheric $CO_2$ growth rate and the decadal trend of the multi-model
average of the ocean carbon sink breaks down if the ocean carbon sink is larger than 4.5 Pg C yr$^{-1}$ (Figure 3, $r^2$ starts to reduce if years with an ocean carbon sink larger than 4.5 Pg C yr$^{-1}$ are included). The breakdown likely occurs because climate change and associated ocean heat uptake and circulation changes become so large that effects on the natural carbon sink reduce the trend in the ocean carbon sink substantially enough. Thus, it is not the carbon uptake of 4.5 Pg C yr$^{-1}$ itself





that causes the breakdown of the relationship but the combined impact of an increasing Revelle factor (Revelle and Suess, 1957) and climate change (Joos et al., 1999; McNeil and Matear, 2013; Frölicher et al., 2015). In SSPs from CMIP6, the combined impact is large enough to affect the here identified relationship when the ocean carbon sink is around 4.5 Pg C yr⁻¹. The breakdown of the relationship also implies that the decadal trends in the ocean carbon sink cannot exceed $0.78 \pm 0.10$ Pg C yr⁻¹ dec⁻¹ (the uncertainty is the 1-$\sigma$ standard deviation across the ESM ensemble in the decade when the multi-model mean decadal trend is largest). Thus, if the ocean carbon sink is below 4.5 Pg C yr⁻¹ and if its magnitude 10 years ago and the change in the decadal trends of atmospheric $CO_2$ between the last two decades (20 to 10 years ago and 10 years to now) is known, the absolute ocean carbon sink this year can be determined (Fig 3b).

### 3.3.2 Regional relationships

The relationship between changes in the atmospheric $CO_2$ growth rate and the global ocean carbon sink holds in all large five ocean basins (Fig. 4a-j) as it has also done from 1980 to 2018 (McKinley et al., 2020). The correlation coefficient is larger than 0.84 in the Atlantic, Pacific, Indian, and Southern Ocean. Only the Arctic Ocean has a smaller correlation coefficient of 0.66.

In the Arctic Ocean, the carbon sink has been shown to be already substantially more affected by climate change than in any other ocean basin (Yasunaka et al., 2023). In the future, when sea ice will disappear and the Arctic will continue to warm faster than any other region, the importance of climate change for the Arctic Ocean carbon sink will likely remain relatively large, for example through freshening (Terhaar et al., 2021a) and a change in the seasonal cycle of $p$CO₂ (Orr et al., 2022), and hence reduce the importance of changes in the atmospheric $CO_2$ for trends in the ocean carbon sink.

In the Southern Ocean, the simulated trends in the ocean carbon also slightly differ from the expected trends based on changes in trends of atmospheric $CO_2$ in three brief periods (Fig. 4i). From 1995 to 2005 over the historical period and from 2030 to 2050 under SSP1-2.6, the decadal trend in the ocean carbon sink is larger than expected, whereas it is smaller than expected from 2080 to 2100 under SSP1-2.6. The differences under SSP-1.2.6 are even visible for the global carbon sink (Fig 3a). As the difference is occurring in the multi-model mean, it appears to be a forced response and not a response that is linked to the particular state of the climate in one of the models. The time periods where the differences are visible globally (2030-2050 and 2080 to 2100 under SSP1-2.6) are the times when the growth in atmospheric $CO_2$ stops and when it starts to decrease in that scenario (Fig 1c). As the atmospheric $CO_2$ growth rate changes quickly in these periods (Fig 2a), first by changing into a decreasing phase and then transitioning into a stabilizing phase. It appears that a fast transition of the trend change in atmospheric $CO_2$ temporarily leads to differences in the expected relationship. If the trend change in atmospheric $CO_2$ decreases fast, the trend in ocean carbon sink remains larger than expected and if the trend change in atmospheric $CO_2$ increases fast, the trend in ocean carbon sink remains smaller than expected. However, the drivers behind the divergence



from the expected decadal trend of the multi-model mean in from 1995 to 2005 in the Southern Ocean remain unclear and should be analysed in future research.

Despite these small differences, the overall relationship between changes in decadal trends in the atmospheric $CO_2$ and decadal trends in the local and global ocean carbon sink is very strong ($r^2>0.84$, apart from the Arctic Ocean) and demonstrates how atmospheric $CO_2$ is the main driver of the externally forced decadal trends of the ocean carbon sink.




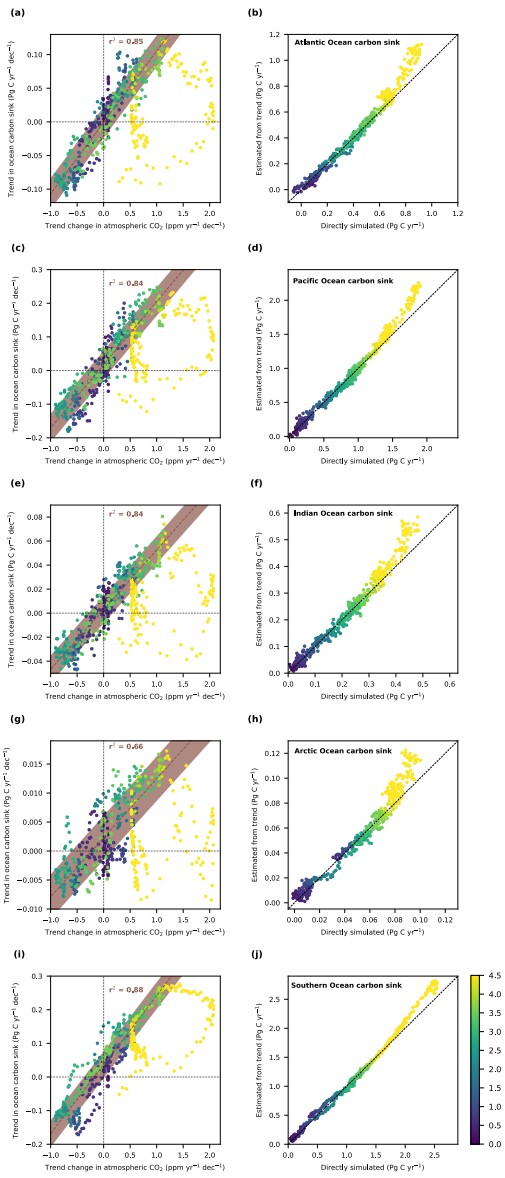

**Figure 4: The relationship between changes in the atmospheric CO₂ growth rate and decadal trends of the ocean carbon sink in the five major ocean basins.** Decadal trends in the ocean carbon sink in the **(a)** Atlantic Ocean, **(c)** Pacific Ocean, **(e)** Indian Ocean, **(g)** Arctic Ocean, and **(i)** Southern Ocean compared to changes in decadal trends in atmospheric CO₂, which represent the decadal averaged growth rate of atmospheric CO₂. The dark blue to yellow circles without a surrounding black line show multi-model averages for all years of the historical period from 1850 to 2014 and for all years from 2015 to 2100 for all four SSPs. The brown line shows a linear fit for all years when the global ocean carbon sink is smaller than 4.5 Pg C yr⁻¹ and the brown shading is the 1-σ projection uncertainty. The dots with black lines around them show values from $p$CO₂ products (pink) and GOBMs (orange) from the Global Carbon Budget 2023 (Friedlingstein et al., 2023)for the three decades between 1990 and 2020. The simulated ocean carbon sink in the **(b)** Atlantic Ocean, **(d)** Pacific Ocean, **(f)** Indian Ocean, **(h)** Arctic Ocean, and **(j)** Southern Ocean in comparison to the expected ocean carbon sink based on the respective relationships in **(a, c, e, g, i)** and the prescribed trend change in atmospheric CO₂ in the simulations.





## 4 The importance of climate variability on decadal trends on the ocean carbon sink

Internal climate and ocean variability in ESMs reduces the strength of the relationship between changes in decadal trends in the atmospheric $CO_2$ and decadal trends in the ocean carbon sink. To quantify the importance of climate variability, I calculated the relationship between changes in decadal trends in the atmospheric $CO_2$ and decadal trends in the ocean carbon sink not for the multi-model mean but for the individual ESMs. When calculating the linear fit over the results from the individual ESMs, the correlation factor only slightly reduces from $r^2=0.91$ to $r^2=0.80$ (Fig 5). The 1-σ prediction interval

around the linear fit is 0.16 Pg C yr$^{-1}$ dec$^{-1}$, meaning that 68% of all trends will be within ±0.16 Pg C yr$^{-1}$ dec$^{-1}$ of the predicted trend based on decadal trends in the atmospheric $CO_2$, 95 % will be within ±0.31 Pg C yr$^{-1}$ dec$^{-1}$ of the predicted trend, and virtually all trends (99.7%) will be within ±0.47 Pg C yr$^{-1}$ dec$^{-1}$ of the predicted trend. The largest simulated trend in the ocean carbon sink in one of the ESMs is 0.96 Pg C yr$^{-1}$ dec$^{-1}$. This is within the 2-σ range of the largest trend as diagnosed by the multi-model mean 0.78±0.10 Pg C yr$^{-1}$ dec$^{-1}$.


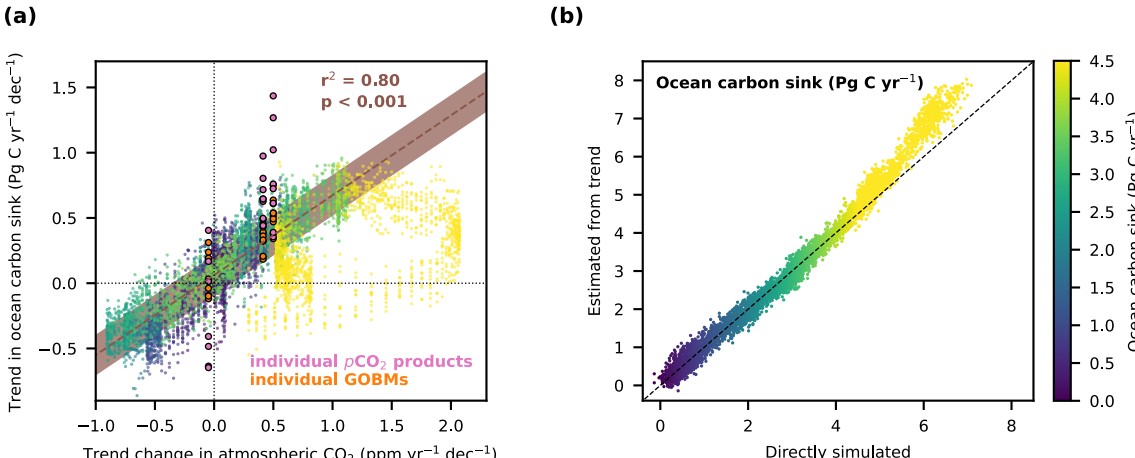

**Figure 5: The relationship between changes in the atmospheric $CO_2$ growth rate and decadal trends of the global ocean carbon sink for individual ESMs. (a)** Decadal trends in the ocean carbon sink for all ESMs individually compared to changes in decadal trends in atmospheric $CO_2$, which represent the decadal averaged growth rate of atmospheric $CO_2$. The dark blue to yellow circles without a
surrounding black line show multi-model averages for all years of the historical period from 1850 to 2014 and for all years from 2015 to 2100 for all four SSPs. The brown line shows a linear fit for all years when the global ocean carbon sink is smaller than 4.5 Pg C yr$^{-1}$ and the brown shading is the 1-σ projection uncertainty. The dots with black lines around them show values from individual $pCO_2$ products (pink) and GOBMs (orange) from the Global Carbon Budget 2023 (Friedlingstein et al., 2023) for the three decades between 1990 and 2020. **(b)** The simulated ocean carbon sink in comparison to the expected ocean carbon sink based on the relationship in **(a)** and the
prescribed trend change in atmospheric $CO_2$ in the simulations.

The range of simulated trends in ocean carbon sink with different internal climate variability encompasses the ocean carbon sink trend estimates of GOBMs from the Global Carbon Budget 2023 (Friedlingstein et al., 2023) but the trend estimates of





the $pCO_2$ products exceed the range that is simulated by ESMs. For the decade from 1990 to 1999, 7 out of 10 GOBMs fall

within the ±1-σ range and the remaining 3 GOBMs fall within the ±2-σ range. In comparison, only 3 out of 8 $pCO_2$ products

fall within the ±1-σ range, 2 $pCO_2$ products fall within the ±3-σ range, 1 $pCO_2$ product falls within the ±4-σ range, 2 $pCO_2$

products fall within the ±5-σ range. For the decade from 2000 to 2009, 5 GOBMs fall within the ±1-σ range, 4 GOBMs fall

within the ±2-σ range, and the GOBM falls within the ±3-σ range. In comparison, only 2 $pCO_2$ products fall within the ±1-σ

range, 1 $pCO_2$ product falls within the ±2-σ range, 2 $pCO_2$ products fall within the ±3-σ range, 1 $pCO_2$ products fall within

the ±5-σ range, 1 $pCO_2$ products fall within the ±6-σ range, and 1 $pCO_2$ product falls within the ±7-σ range. For the decade

from 2010 to 2019, 9 GOBMs fall within the ±1-σ range and the remaining one falls within the ±2-σ range. In comparison,

only 1 $pCO_2$ products fall within the ±1-σ range, 2 $pCO_2$ products fall within the ±2-σ range, 3 $pCO_2$ products fall within the

±3-σ range, 1 $pCO_2$ product falls within the ±4-σ range, and 1 $pCO_2$ product falls within the ±5-σ range. In general, the

$pCO_2$ product estimates of the decadal trends are not randomly distributed across the possible range that the ESMs suggest.

Instead, $pCO_2$ products systematically overestimate the magnitude of the respective trends that is suggested by ESMs, i.e., a

too small negative trend in the 1990s and a too high positive trend in the 2000s and 2010s.





## 5 Imprint of climate change and ocean heat uptake on the ocean carbon sink

In addition to atmospheric $CO_2$ and internal climate variability, climate change and ocean heat uptake also affects the ocean

carbon sink and potentially its decadal trends. The ocean heat uptake, for example, causes to changes in the ocean circulation such as stratification and outgassing of natural carbon from the ocean due to increasing temperatures and reduced solubility (Fig. 6). Across the five ESMs that performed the simulations to quantify the effect of ocean heat uptake on the natural carbon in the ocean (see Methods), the loss of natural carbon from the ocean to the atmosphere is related to the ocean heat uptake via a 2nd degree polynomic function under strong radiative forcing (SSP5-8.5) (Fig. 6a). Although annual variability

hides part of this relationship, the relationship emerges strongly for decadal averages (Fig. 6b).

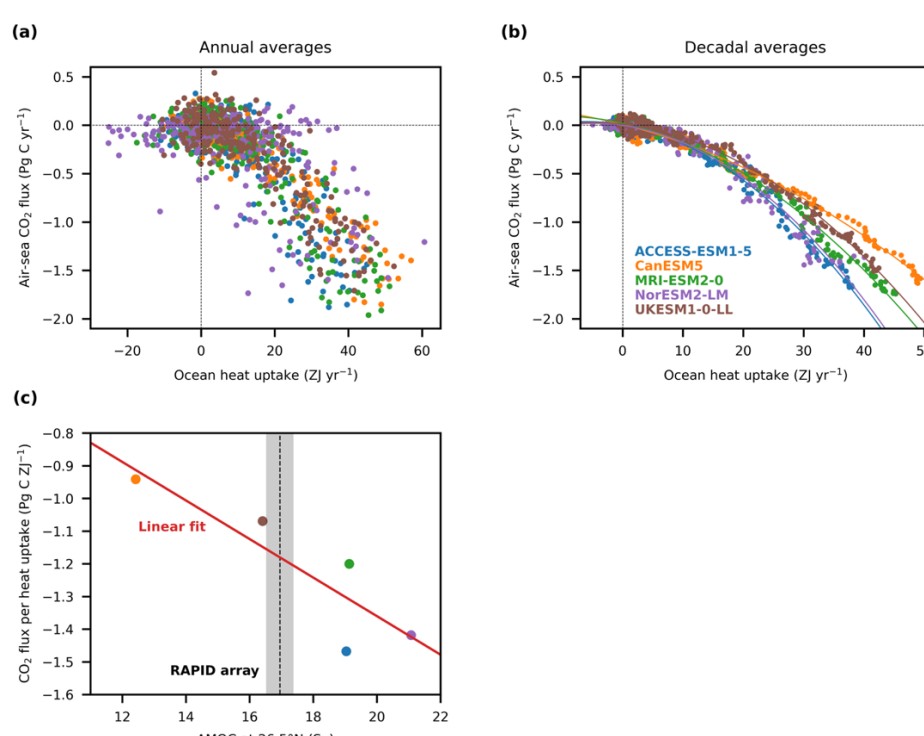

**Figure 6: The relationship between natural carbon loss and ocean heat uptake and its link to the Atlantic Meridional Overturning Circulation. (a)** Annually and **(b)** decadal averaged air-sea $CO_2$ flux solely caused by ocean heat uptake (for details see Methods) vs the

annually averaged ocean heat uptake in five different ESMs (ACCESS-ESM1-5 in blue, CanESM5 in orange, MRI-ESM2-0 in green, NorESM2-LM in purple, and UKESM1-0-LL in brown). A 2nd degree polynomic function (coloured lines) was fitted over the decadal averaged values of each ESM. **(c)** The $CO_2$ flux per ocean heat uptake, calculated for each model from the fitted 2nd degree polynomic function at an ocean heat uptake at 35 ZJ yr$^{-1}$, against the Atlantic Meridional Overturning Circulation (AMOC) at 26.5°N calculated in each ESM from 2004 to 2018 (historical plus SSP5-8.5 simulations).




While each of the five ESMs suggests that the loss of natural carbon from the ocean to the atmosphere is related to the ocean heat uptake via a 2$^{nd}$ degree polynomic function, the amount of carbon loss due per ocean heat uptake varies across ESMs (Fig 5b). The main reason for differences likely the different changes in ocean circulation and stratification due to ocean heat uptake in each ESM. One of the parts of the ocean overturning circulation that is expected to change strongly with climate

change and ocean warming in ESMs is the Atlantic Meridional Overturning Circulation (Weijer et al., 2020). Across a large ensembles of ESMs from CMIP6, it has been shown that ESMs with an already stronger Atlantic Meridional Overturning Circulation also show a stronger reduction in the Atlantic Meridional Overturning Circulation (Weijer et al., 2020). The larger overturning reduction thus causes the models with a higher Atlantic Meridional Overturning Circulation to lose more natural carbon loss for the same heat uptake (Fig. 5c). Based on this linear relationship, it would be possible to constrain the

loss of carbon per heat uptake with observations of the present-day Atlantic Meridional Overturning Circulation. However, using only five ESMs to quantify a linear relationship is likely not yielding a robust relationship so that I abstain from constraining the loss of carbon per heat uptake. Nevertheless, the observed Atlantic Meridional Overturning Circulation at 26.5°N is close to the average of the simulated Atlantic Meridional Overturning Circulation at 26.5°N in ESMs suggesting that the multi-model average sensitivity of air-sea $CO_2$ fluxes to heat uptake are a good approximation of the real-world

sensitivity.

Unfortunately, CMIP6 only provides simulations that allow to quantify the ocean natural carbon sink response to ocean warming for SSP5-8.5 and not for other scenarios where the ocean warming slows down or even stabilizes. Thus, it remains impossible for now to quantify the effect of ocean heat uptake for other scenarios and to test if the here identified

relationship is robust. However, as differences in the decline of the Atlantic Meridional Overturning Circulation are similar across all these scenarios although the ocean heat uptake is much smaller in the low emission scenarios (Weijer et al., 2020), the sensitivity of carbon loss to heat uptake might be larger in low-emission scenarios. As such, changes in the ocean heat uptake and its trend might well cause changes in the anthropogenic ocean carbon sink via the outgassing of marine natural carbon pool. Although these changes are likely small as decadal averaged ocean heat uptake does not change quickly, these

changes might still be partly responsible for differences between the decadal trend of the ocean carbon sink that were expected based on changes in trend of atmospheric $CO_2$ and the simulated ocean carbon sink, especially those in SSP1-2.6 globally (Fig. 3) and in the Southern Ocean (Fig. 4i). To verify this hypothesis, CMIP simulations that allow to quantify the ocean natural carbon sink response to ocean warming would have to be made for other scenarios than SSP5-8.5.



## 6 Potential caveats and limitations

The strong dependence of decadal trends in ocean carbon sinks on the change of the atmospheric $CO_2$ growth rate from one decade to the other was here identified across an ensemble of state-of-the-art ESMs from CMIP6. The robustness of this relationship depends on the model's ability to represent the internal climate variability and might also be biased if the entire model ensemble is biased, for example due to relatively coarse resolution or a common unrealistic representation of the physics or biogeochemistry.

If, for example, the internal climate variability on decadal timescales was underestimated by the here-used ESMs, the predictability of the decadal trends in the ocean carbon sink by changes in the growth rate of $CO_2$ would be overestimated. A prerequisite for ESMs to simulate the contribution of the natural variability to decadal trends of the ocean carbon sink is that they also simulate the size of the decadal trends of internal climate modes that are known to affect the variability of the ocean carbon sink most. The major climate modes that are known to influence the decadal variability of the ocean carbon sink are the Niño 3.4 index (Feely et al., 1999; Ishii et al., 2014; McKinley et al., 2004), Atlantic Multi-decadal Oscillation (Breeden and McKinley, 2016; Keppler et al., 2023), and Marshall Southern Annular Mode (Le Quéré et al., 2007; Gruber et al., 2019b; Landschützer et al., 2015; Lovenduski et al., 2008; Thompson and Solomon, 2002; Lenton and Matear, 2007; Hauck et al., 2013). The decadal trends of the Niño 3.4 index in ESMs is 17 (-19 to 53) % larger than the decadal trends of the observation-based estimates of the Niño 3.4 index (the numbers in parenthesis indicates the standard deviation across ESMs) (Fig 7b), the decadal trends of the Atlantic Multi-decadal Oscillation in ESMs are 13 (-7 to 33) % larger than the decadal trends in the observation-based estimate  (Fig 7c), and the decadal trends of the Marshall Southern Annular Mode in ESMs are 52 (30 to 75) % larger than the decadal trends in the observation-based estimate (Marshall, 2003) (Fig 7d). The relatively large decadal trends of climate modes in ESMs suggest that the ESMs are indeed capable of simulating the internal climate variability on decadal timescales. Thus, there is no indication that the decadal variability of the ocean carbon sink in ESMs (Fig 7a) might be too small because of a too small internal climate variability in ESMs as previously hypothesized (Gruber et al., 2023).



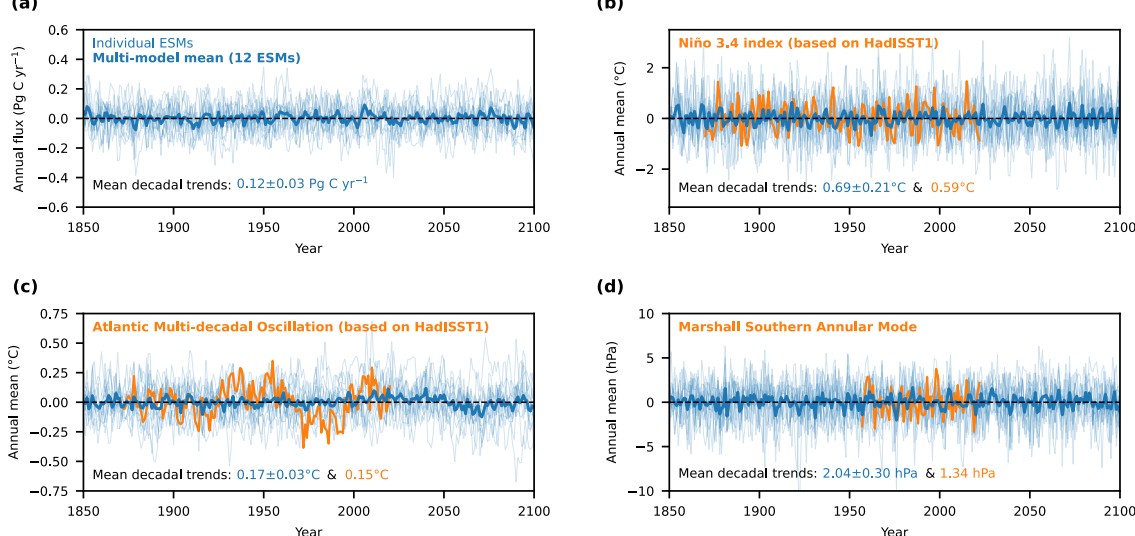


**Figure 7: Timeseries and decadal trends of the ocean carbon sink and climate modes in earth system models compared to observations. (a)** The globally integrated annual air-sea $CO_2$ flux in the pre-industrial control simulations for 12 ESMs (thin blue lines) and the multi-model average (thick blue line). The same is shown for **(b)** the Niño 3.4 index, **(c)** the Atlantic Multi-decadal Oscillation, and **(d)** the Marshall Southern Annular Mode. For the three climate modes, observation-based estimates are shown based on HadISST1 for **(b, c)** and based on Marshall (2003) for **(d)**. The decadal trends of these observation-based estimates (orange numbers) are compared to the decadal trends of ESM estimates (blue numbers indicating average and standard deviation across the ESM ensemble).

However, in addition to physical climate-driven variability, which is the dominant driver of variability of the ocean carbon
sink (Doney et al., 2009), there is also biology- and biogeochemical-driven climate-related variability in the air-sea $CO_2$
fluxes (Ostle et al., 2022; Doney et al., 2009; Keller et al., 2012) for example due to changes in net primary production or
remineralization caused by changes in nutrient supply, temperature, and oxygen. Over the North Atlantic, it has been shown
that biogeochemical variability is also strongly influenced by climate modes, such as the Atlantic Multi-decadal Oscillation
(Ostle et al., 2022). Nevertheless, GOBMs suggest that the influence of physical variability exceeds the influence of
biogeochemical variability (Doney et al., 2009; DeVries et al., 2023). Despite different representations of the
biogeochemistry and biology across the models from RECCAP2 (DeVries et al., 2023; Rodgers et al., 2023), they all
simulate a similar inter-annual and decadal variability and trends in the ocean carbon sink (DeVries et al., 2023; Terhaar et
al., in press) as they are forced with historical atmospheric reanalyses products and share the same internal climate modes.
Although the similarity of all GOBMs when forced with historical reanalysis strongly suggest that the physical impact on
decadal variability exceeds the biogeochemical impact, detailed regional analyses of the biogeochemical-driven climate-
related variability in the air-sea $CO_2$ fluxes (Ostle et al., 2022; Keller et al., 2015), which exceed the scope of this
manuscript, are necessary. Overall, the dominance of physical variability over biogeochemical variability and the larger



decadal trends of climate modes in ESMs than in the real world suggest that the ESMs do not underestimate the natural variability of the ocean carbon sink.


Although the here used ESMs from CMIP6 simulate even larger decadal trends of important climate modes, they might still underestimate decadal trends of the ocean carbon sink driven by climate variability because of their resolution that has increased over the past decades but is still too coarse to explicitly resolve mesoscale ocean eddies. Higher resolved ESMs are still computationally too expensive to be run within the CMIP framework with sufficiently long spin-ups that are necessary

for these models to be in equilibrium (Séférian et al., 2016; Gupta et al., 2013). In a few studies with less simulations than required for CMIP, higher-resolved ocean models have been shown to affect the ocean carbon sink and physics and their variability (Lachkar et al., 2007, 2009; Dufour et al., 2015; Griffies et al., 2015). While it remains impossible to evaluate the effect of higher resolution over a large ensemble of ESMs, as such an ensemble does not exist yet, I tested the here-identified relationship with the highest-resolved earth system model within CMIP6, GFDL-CM4 (Held et al., 2019), which has a

horizontal resolution of 0.25° that allows to resolve eddies in tropical and subtropical oceans but still has to parametrize some eddy activity in subpolar and polar oceans. GFDL-CM4 had not been included in the overall analyses as it did not provide simulations under SSP1-2.6 and SSP3-7.0, presumably because of its large computational costs. The trends in the ocean carbon sink in GFDL-CM4 lie mostly within $\pm 1\sigma$ of the relationship between changes in the atmospheric $CO_2$ growth rate and trends in the ocean carbon sink, with only a few decades being in the $\pm 2\sigma$ range (Figure 8). As for the other ESMs,

the relationship in GFDL-CM4 only holds if the ocean carbon sink remains below 4.5 Pg C yr$^{-1}$. Although a potential change in this relationship at an even higher resolution cannot be excluded with certainty until simulations with higher resolution are performed, the robustness of the relationship even for higher-resolved ESMs such as GFDL-CM4 gives no indication that the relationship will not hold at even higher resolution.


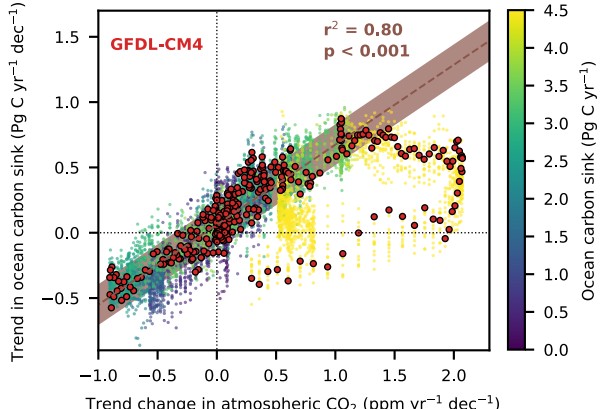



**Figure 8: The relationship between changes in the atmospheric CO₂ growth rate and decadal trends of the global ocean carbon sink in a high-resolution model. (a)** Decadal trends of the multi-model mean ocean carbon sink compared to changes in decadal trends in atmospheric $CO_2$, which represent the decadal averaged growth rate of atmospheric $CO_2$. The dark blue to yellow circles without a surrounding black line show multi-model averages for all years of the historical period from 1850 to 2014 and for all years from 2015 to 2100 for all four SSPs. The brown line shows a linear fit for all years when the global ocean carbon sink is smaller than 4.5 Pg C yr$^{-1}$ and the brown shading is the 1-σ projection uncertainty. The red dots with black lines around them show values for the high-resolution ESM GFDL-CM4 (Held et al., 2019) under the historical simulation and SSP2-4.5 and SSP5-8.5.



## 7 Discussion and Conclusion

The analysis with ESM suggests that changes in the atmospheric growth rate of $CO_2$ can indeed explain most of the decadal trends of the ocean carbon sink, as previously proposed by McKinley et al. (2020). ESMs support the hypothesis by McKinley et al. (2020) that the weak decadal trend in the 1990s and the stronger trend in the 2000s is mainly driven by

changes in the atmospheric $CO_2$ growth rate (Figures 3 and 5). The importance of atmospheric $CO_2$ extends over all ocean basins, as also previously shown by McKinley et al. (2020). While McKinley et al. (2020) have focused on the last decades and suggested that the trends in the ocean carbon sink depends on differences on the atmospheric growth rate of $CO_2$ compared to the long-term trend of the growth rate, I could show that it is the change in the growth rate compared to the previous decade that drives the trends of the ocean carbon sink. Moreover, this analysis here extends the timeline of previous

analysis and shows how atmospheric $CO_2$ drives trends in the ocean carbon sink on a range of different future scenarios, from high-mitigation low emission scenarios to high-emission scenarios. In addition, the use of ESMs allowed to quantify the link between changes in the atmospheric $CO_2$ growth rate and decadal trends of the ocean carbon, which allows to better separate the effect of internal and external forcing of past decadal trends of the ocean carbon sink. However, if atmospheric $CO_2$ rises too high and the impact of climate change on the ocean carbon sink increases, atmospheric $CO_2$ is not anymore the

dominant driver of trends of the ocean carbon sink due to changes in the buffer factor and ocean ventilation (Revelle and Suess, 1957; Heinze et al., 2015; Joos et al., 1999; McNeil and Matear, 2013; Frölicher et al., 2015).

Although atmospheric $CO_2$ is here shown to be the main driver of the decadal trends in the ocean carbon sink, climate variability also plays an important role for the decadal trends of the ocean carbon sink. With a standard-deviation of $\pm 0.16$

Pg C yr$^{-1}$ across all ESMs, climate variability drives 17% of the trend in the ocean carbon sink when the growth rate of atmospheric $CO_2$ is largest and drives all changes in carbon trends when the change in the growth rate of atmospheric $CO_2$ is zero. Known drivers of this internal climate variability are for example El Niño (Feely et al., 1999; Ishii et al., 2014; McKinley et al., 2004), the Atlantic Multi-decadal Oscillation (Breeden and McKinley, 2016; Keppler et al., 2023), changes in the overturning circulation (DeVries et al., 2017), and changes in the Southern Annual Mode liked to changes in Southern

Ocean winds (Le Quéré et al., 2007; Keppler and Landschützer, 2019; Landschützer et al., 2015) and stronger consequent upwelling of older waters (Lovenduski et al., 2008, 2007), as well as changes in the Southern Ocean stratification (Gruber et al., 2019b). Across the here-used ESMs, the variability of decadal trends is highest in the Southern Ocean, followed by the tropical regions, and again followed by the Northern subpolar gyres (Figure 9); confirming that the decadal trends of the ocean carbon sink are indeed most variable due to internal climate variability in the regions where they are expected based

on the previous studies mentioned above. ESMs simulate decadal trends of important climate modes that are even larger than their observation-based counterparts, which suggests that ESMs also capture the climate-driven decadal trends of the ocean carbon sink. As ESMs slightly overestimate decadal trends of important climate modes and still suggest that changes in the





atmospheric growth rate are the dominant drivers of decadal trends of the ocean carbon sink, climate variability and associated changes in ocean circulation appear to not be the first order driver of decadal trends of the ocean carbon sink over

the last decades as previously suggested (Landschützer et al., 2015; DeVries et al., 2017, 2019).

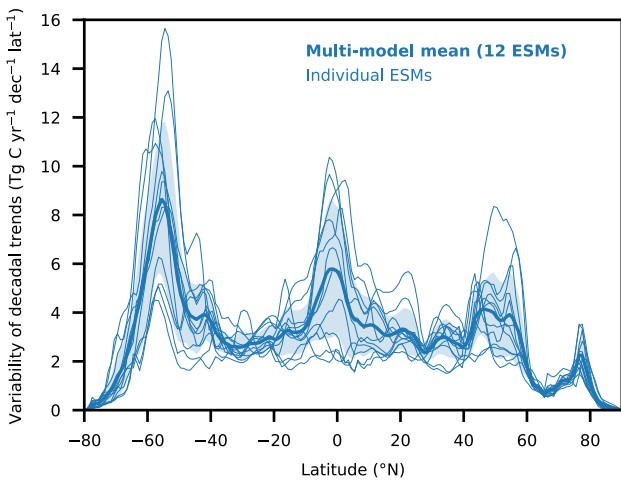

**Figure 9: Variability of the decadal trends of the zonally integrated ocean carbon sink in earth system models.** The variability of the
zonally integrated ocean carbon sink across the 251 years of the pre-industrial control simulation in each of the 12 ESMs (Table 1) (thin
blue lines). In addition, the multi-model mean (thick blue line) and the 1-σ standard deviation across all 12 ESMs (blue shading) is shown.

The here presented results have implications for previous estimates of the ocean carbon sink, especially those from $p\mathrm{CO_2}$
products that suggested very strong decadal trends of the ocean carbon sink (Landschützer et al., 2015; Gruber et al., 2019b,
2023). The trend estimates of the ocean carbon sink by $p\mathrm{CO_2}$ products is larger than the likely trends based on the here-
identified relationship between changes in the atmospheric $\mathrm{CO_2}$ growth rate and the decadal trends of the ocean carbon sink.
Thus, $p\mathrm{CO_2}$ products either overestimate decadal trends of the ocean carbon sink or ESMs underestimate these trends. In
each decade from 1990 to 2019, there are five out of eight $p\mathrm{CO_2}$ products from the Global Carbon Budget (Friedlingstein et
al., 2023) that estimate decadal trends that are outside of the 2-σ range that is estimated based on ESMs here (Figure 5),
giving these results a likelihood of less than 5% to occur if the ESM results are indeed robust. Some $p\mathrm{CO_2}$ products estimate
trends that are within the 5-σ, 6-σ, and 7-σ ranges that corresponds to events that occur once every 4776 years (5-σ), once
every 1.38 million years (6-σ), and once every 1.07 billion years (7-σ). While it is already extremely unlikely that decadal
trends in all three decades from 1990 to 2020 lie outside the 2-σ range, the estimates within the 5-σ to 7-σ range are virtually
impossible based on the ESM-derived range. Here, I have demonstrated that ESMs are capable of simulating the size of
decadal trends of important climate modes that have strong impact on the variability of the ocean carbon sink (Figure 7) and



that higher resolution does not alter the identified relationship (Figure 8). While this analysis does not guarantee that ESMs do not underestimate the decadal trends in the ocean carbon sink, it suggests that ESMs can simulate the size of the variability of the ocean carbon sink. This conclusion also challenges earlier findings that GOBMs might underestimate decadal trends of the ocean carbon sink (DeVries et al., 2019). Other studies (Gloege et al., 2021; Hauck et al., 2023) support

the hypothesis of an overestimation of decadal trends in the ocean carbon sink by $pCO_2$ products. Hauck et al. (2023) have recently demonstrated with one GOBM and two $pCO_2$ products that sampling biases of $pCO_2$ have caused trends in the ocean carbon sink to be overestimated. A similar finding has been made previously, when data from one GOBM, which was sampled in the same way as the real-world ocean was sampled, was extrapolated by one $pCO_2$ products to reconstruct the ocean carbon sink. This so-reconstructed ocean carbon sink by the $pCO_2$ products also had a larger variability than the

directly simulated ocean carbon sink by the GOBM (Gloege et al., 2021). Thus, it appears that most of the $pCO_2$ products have and still overestimate decadal trends of the global ocean carbon sink. Therefore, estimates of the variability and size of the flux of natural carbon based on the difference of the total air-sea $CO_2$ flux from $pCO_2$ products and the change of interior ocean anthropogenic carbon (Müller et al., 2023; Gruber et al., 2019a), defined in this special case only as the additional carbon from increasing atmospheric $CO_2$ and not from climate change, are also likely too large.


The here found dependence of the ocean carbon sink on atmospheric $CO_2$ also has implications for studies that extrapolate present-day observation-based estimates of the ocean carbon sink back in time over the entire historical period to estimate a cumulative ocean carbon sink since the beginning of the pre-industrial revolution using the difference of atmospheric $CO_2$ since pre-industrial times as a scaling factor (Gruber et al., 2009; Mikaloff Fletcher et al., 2006). While this scaling works

approximately for most of the historical period, it breaks down from 1920 to 1960 and in the 1990s (Figure 1). In addition, such estimates might be highly sensitive to the year for which the ocean carbon sink was estimated based on observations. If that year falls in one of these anomalous periods, as the year 1995 in Mikaloff Fletcher et al. (2006), the scaling might be biased low or high. Therefore, these extrapolations of present-day fluxes over the historical period should be used with caution or with a slightly more complex extrapolation method that takes the change in the atmospheric $CO_2$ growth rate into

account.

The importance for changes of the atmospheric $CO_2$ growth rate for the trends of the global ocean carbon sink also affects our understanding of the uncertainty of the ocean carbon sink and the role of internal variability in the future. Previous studies have used CMIP simulations with prescribed atmospheric $CO_2$ to quantify the importance of internal variability for

the uncertainty of the projections of the ocean carbon sink in comparison to the importance of model and scenario uncertainty (Gooya et al., 2023; Lovenduski et al., 2016; Schlunegger et al., 2020). As these prescribed atmospheric $CO_2$ timeseries in CMIP simulations are much smoother than observed atmospheric $CO_2$ timeseries (Fig. 1), changes in the atmospheric $CO_2$ growth rate are also much smaller. Thus, these concentration driven CMIP SSPs suppress the internal variability of the atmospheric $CO_2$ growth rate caused by variabilities in atmospheric temperature, precipitation, El Niño, and



volcanic eruptions (Keeling et al., 1995; Kuo et al., 1990; Raupach et al., 2008; Zeng et al., 2005; Bacastow, 1976; Yang and Wang, 2000). The suppressed variability of the atmospheric $CO_2$ growth rate in concentration driven SSPs also suppresses the variability of the ocean carbon sink in the future, leading to an underestimation of the importance of the internal variability in ESMs for the overall uncertainty of ocean carbon sink projections over the 21st century (Gooya et al., 2023; Lovenduski et al., 2016; Schlunegger et al., 2020). This underestimation of the variability of the ocean carbon sink due to

prescribed atmospheric $CO_2$ can be avoided if ESMs were run in an emission-driven mode that automatically introduces a strong variability of the of the atmospheric $CO_2$ growth rate, as in the model intercomparison project using the Adaptive Emission Reduction Approach (Terhaar et al., 2022a; Silvy et al., 2024).

While changes in the atmospheric $CO_2$ growth rate and ocean heat uptake might allow to estimate changes in the decadal

variability of the ocean carbon sink, it remains still unknown how climate variability and individual modes can be used to predict inter-annual variability of the near-term ocean carbon sink (Lovenduski et al., 2019). In addition, other external forcings, such as volcanic eruptions, are an important factor to the inter-annual variability of the ocean carbon sink but also contribute to decadal trends (McKinley et al., 2017; Fay et al., 2023; Frölicher et al., 2011, 2013).

The influence of changes in the atmospheric $CO_2$ growth rate on the ocean carbon sink also has profound implications on the near-term future of the ocean carbon sink. With less strongly increasing or even peaking carbon emissions the atmospheric $CO_2$ growth rate will also peak and potentially decline. The growth rate of atmospheric $CO_2$ in Mauna Loa has shown a robust negative trend since 2016 and the last time that the growth rate of atmospheric $CO_2$ has been as small as in 2022 was the year 2008 (https://gml.noaa.gov/ccgg/trends/gr.html). If this change from a rise of the atmospheric $CO_2$ growth rate

towards a decline of the atmospheric $CO_2$ growth rate continues, the forced trend in the ocean carbon sink will also be negative. In addition, GOBMs and $pCO_2$ products suggest that the internal climate variability has led to particularly positive trends of the ocean carbon sink in the 2000s and 2010s (Figs 3 & 5, orange and pink dots lie above the derived relationship). This climate variability will eventually reverse at some point and lead to a larger decline of the ocean carbon sink. Furthermore, ocean heat uptake is projected to increase over the next decade or two independent of the chosen future

pathway, also leading to stronger outgassing of carbon from the ocean in the near future and negative trends in the ocean carbon sink (Fig. 6). Although CMIP6 ESMs tend to simulate a smaller ocean heat content over the last two decades (Lyu et al., 2021), this might not be an overestimation by heat uptake of the ESMs but an especially low uptake due to climate variability. Indeed, the recent strong increase in ocean heat content and sea surface temperatures in 2023 (Cheng et al., 2024) might be the beginning of a shift from a period of low heat uptake due to climate variability to a period of high ocean heat

uptake. Together, the decreasing atmospheric $CO_2$ growth rate, the potential change in internal climate variability, and the increased ocean heat uptake will likely cause a substantial negative trend of the ocean carbon sink over the next decade. If, however, emissions and atmospheric $CO_2$ will rise, the continuous increase in the atmospheric $CO_2$ growth rate will cause the ocean carbon sink to increase as well.



Overall, this study demonstrate how ESMs can be used to better understand the past and future of the ocean carbon sink and drivers of its variability. They hence provide not only a valuable addition to $p$CO$_2$ products and GOBMs, but also a unique tool to statistically assess uncertainties and drivers of variability, also potentially in the interior ocean (Müller et al., 2023). The robustness in these results is further corroborated by their capability to simulate the size of decadal trends of important climate modes.




**Data availability**

The Earth system model output used in this study is available via the Earth System Grid Federation (https://esgf-node.ipsl.upmc.fr/projects/esgf-ipsl/, last access: 1 June 2022).

**Competing interests**

The author has declared that he has no competing interests.

**Acknowledgments**

First, I want to thank Thomas L. Frölicher, Fortunat Joos, Roland Séférian, and Jens D. Müller for their helpful comments on the manuscript and during discussions. Furthermore, I acknowledge funding from the Woods Hole Oceanographic Institution

Postdoctoral Scholar Program, and the Swiss National Science Foundation under grant # PZ00P2_209044 (ArcticECO).




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
