# Peer review of "Drivers of decadal trends of the ocean carbon sink in the past, present, and future in Earth system models"

_EGUsphere, 2024_

## Author Comment (AC1)

I am deeply thankful to the anonymous reviewer for their constructive and very helpful evaluation of the manuscript, which has strongly improved its quality. Below, I have addressed the points that were raised by the reviewer point by point. The reviewer's text is shown in black and my responses in blue. Text that will be part of the revised manuscript is shown in italic and blue color.

This manuscript is a nice addition to the long-standing scientific debate about what drives decadal to multi-decadal variability in the ocean carbon sink. The manuscript is comprehensive and mostly well written and clearly structured. The methods and results are well explained and appear robust.

My one truly major comment is regarding the lack of proper statistical analysis. This will have to be added before the manuscript is published. In addition I want to raise several other issues that should be considered and which I think would help improve the manuscript and increase its impact.

**Response:** Thank you for your positive and constructive evaluation of the manuscript.

General:

Occasionally the reading is marred by awkward sentences. The second sentence in the abstract exemplifies this: Observing the ocean carbon sink is not challenging because there are high uncertainties, but the uncertainties are high because observing the ocean is challenging.

**Response:** Thank you for your advice to improve the English. The sentence was adjusted as suggested:

*"Despite the ocean's importance for the carbon cycle and hence the climate, uncertainties of the decadal variability of this carbon sink and the underlying drivers of this decadal variability remain large because observing the ocean carbon sink and detecting anthropogenic changes over time remain challenging."*

Throughout the manuscript the author states that observation-based $pCO_2$ products extrapolate observations. I do not like the use of the word "extrapolate" here. When we extrapolate we extend into an unknown situation, but the observation-based products primarily attempt to interpolate between known situations. It is a bit nit-picky, and probably boils down to semantics, but I'd prefer the term "gap-filling". In my mind that is more comprehensive.

**Response:** I slightly prefer to keep the word "extrapolation" to guarantee consistency with previous literature. Fay et al. (2021) provided a reference for other studies when they created a harmonization 6 such $pCO_2$ products with their developers as co-authors. In this study, they use the word 'extrapolation'. However, I have no strong opinion and am happy to change 'extrapolate' to 'gap-filling' if this is preferred by the reviewer and the editor.

Introduction:

The introduction begins with a statement that the ocean has removed about 25% of all anthropogenic emissions since the onset of the industrial revolution. This is not factually incorrect, but I still find the statement somewhat misleading. Based on Table 8 in Friedlingstein et al. (2023) the cumulative ocean uptake (both since 1750 and since 1850) amounts to approximately 25% of the total emissions (FF+LUC). However, given this statement at the beginning of section 3.9 in the same paper "The cumulative land sink is almost equal to the cumulative land-use emissions (220±70 Gt C), making the global land nearly neutral over the whole 1850–2022 period." we understand that ocean has been more important in storing human-made emissions than the "has taken up around one quarter of all anthropogenic emissions" would indicate. In a paper highlighting the ocean sink I think this is a nuance worth noting in the introduction. But I stress again that the statement is not actually incorrect.

**Response:** As suggested by the reviewer, a sentence was added to the Introduction:

"If land-use change emissions are considered part of the land carbon sink, the land becomes almost neutral and the ocean carbon sink becomes the only major natural carbon sink (Friedlingstein et al., 2023)".

In the introduction (lines 92 onwards) the point is made that differences between observation-based products and GOBM products could be due to the data sparsity. Here it should be noted that it is not the sparsity that is the major problem, but rather the uneven sampling in time and space. Hauck et al. (2023) showed that if observations were regularly spaced the differences largely disappear. This is worth mentioning because it is a problem much more difficult to remedy than just having too few observational data.

**Response:** Following the suggestion by the reviewer, the sentence was changed to:

*"As opposed to the magnitude, the differences in the decadal trends between $pCO_2$ products and GOBMs might be due to uneven sampling of observations in space and in time, e.g., few observations in the 1980s and 1990s and few observations in the southern hemisphere, as demonstrated with a subset of $pCO_2$ products evaluated with output from a GOBM (Hauck et al., 2023) and ESMs (Gloege et al., 2021)."*

Throughout the manuscript it can be difficult to understand what "drivers of the decadal trends" actually mean. My understanding is that this study looks at defining the underlying causes of multi-decadal variability, that is, variability in the decadal trends. I could be mistaken, but this should regardless be stated more clearly in the introduction.

**Response:** Following the suggestion by the reviewer, the respective sentence in the Introduction was extended for clarification:

*"Here, I use an ensemble of 12 ESMs from phase 6 of the Coupled Model Intercomparison Project (CMIP6) (Table 1) to provide a new perspective on potential drivers of the decadal trends of the ocean carbon sink, i.e., the underlying causes of its multi-decadal variability."*

In the introduction several limitations to studies using the observation-based and GOBM products are presented. This is correct and fair, but there are also limitations to using ESMs so a few sentences describing these should be added at the end of the introduction.

**Response:** As suggested by the reviewer, we have added the following text about ESMs to the Introduction:

*"Here, I use an ensemble of 12 ESMs to provide a new perspective on potential drivers of the decadal trends of the ocean carbon sink from phase 6 of the Coupled Model Intercomparison Project (CMIP6) (Table 1). Fully coupled ESMs are another tool to quantify and understand the ocean carbon sinks (e.g., Joos et al., 1999; McNeil and Matear, 2013; Frölicher et al., 2015; Goris et al., 2018; Terhaar et al., 2022b, 2021b). As ESMs are fully coupled and not forced with atmospheric reanalysis data, they do not simulate the same inter-annual internal climate variability as $pCO_2$ products and GOBMs do and their biases of the surface ocean physics and biogeochemistry are thus larger than surface ocean biases of GOBMs (Terhaar et al., 2022b; Terhaar et al., 2024). However, ESMs have distinctive advantages compared to $pCO_2$ products and GOBMs for the analyses of decadal drivers of the ocean carbon sink because (1) they cover a period of 251 years from 1850 to 2100, (2) cover at least four different future scenarios, and (3) they all have a different internal climate state."*

Methods:

In section 2.2 the author describes how the magnitude of the ocean carbon sink in the 12 different ESMs was corrected/adjusted. First, I would think that the magnitude of the ocean carbon sink is related to the model's climate state since it is closely linked to the ocean circulation. So what are the consequences of doing such a correction to the models? Second, it is stated as fact that a "negative bias in the magnitude of the carbon sink also introduces a negative bias in the decadal trends". I do not understand why this will always be the case. This part of the method requires a bit more description, and a bit more discussion.

**Response:** The adjustment allows to bias-correct simulated output based on previously identified biases in the ocean circulation and surface ocean carbonate chemistry (Terhaar et al., 2022b).

Below, I have re-made the main figure from the manuscript without the adjustment. Without the bias-adjustment for each Earth System Model, the strength of the relationship slightly reduces from $r^2$=0.90 to $r^2$=0.83 but remains strong and significant. Hence, the findings are not sensitive to the bias-adjustment, but the bias-adjustment still yields more reliable results as the studies from Goris et al. (2018), Terhaar et al. (2022b) and Terhaar, Goris, et al. (2024) suggest. For clarification, I added the following sentence to the Methods:

*"The adjustment corrects for known biases in the models' circulations and surface ocean carbonate chemistry and hence reduced differences in the overall magnitude of the simulated carbon sink between ESMs (Terhaar et al., 2022b). This reduction in the difference in the magnitude of the carbon sink also reduces differences between the magnitude of trends and slightly improves the relationships found here as it ($r^2$ in Figure 3 would have been 0.83 without*

*adjustment instead of 0.91 with adjustment). Nevertheless, the results would quantitatively and qualitatively almost identical with and without that adjustment. "*

[Figure]

**Figure 3:** The relationship between changes in the atmospheric CO2 growth rate and decadal trends of the global ocean carbon sink for the multi-model mean. **(a)** Decadal trends of the multi-model mean ocean carbon sink compared to changes in decadal trends in atmospheric CO2, which represent the decadal averaged growth rate of atmospheric CO2. The dark blue to yellow circles without a surrounding black line show multi-model averages for all years of the historical period from 1850 to 2014 and for all years from 2015 to 2100 for all four SSPs. All decades over from 1850 to 2100 are shown, i.e., 2000-2009, 2001-2010, 2002-2011, etc. The brown line shows a linear fit for all years when the global ocean carbon sink is smaller than 4.5 Pg C yr-1 and the brown shading is the 1-s projection uncertainty. The dots with black lines around them show values from the respective ensemble means of the pCO2 products (pink) and GOBMs (orange) from the Global Carbon Budget 2023 (Friedlingstein et al., 2023) for the three decades between 1990 and 2020. **(b)** The simulated ocean carbon sink in comparison to the expected ocean carbon sink based on the relationship in **(a)** and the prescribed trend change in atmospheric CO2 in the simulations.

In the introduction the author argues that one of the benefits of using ESMs over other types of data products is the ability to perform robust statistics. I completely agree, but it seems as if no, or very little, statistical analysis has been performed. At least there is no description of such analyses in the methods. This I think is a major weakness of the manuscript as it stands now. At the very least a table with statistics for Figures 3, 4, 5 should be added, but a more thorough statistical approach to the analysis in section 5 and 7 would also be beneficial. I also note here that the author presents a p-value for the regression analysis (Figures 3, 5, 8). The p-value has been a topic of discussion for several years now, and is often misused. At the very least you must state what null hypothesis you are testing. However, given that you have a lot of data points the p-value is perhaps not the most useful metric. Given the amount of data available when using several ESMs I would suggest you try a Bayesian hypothesis testing instead for a more useful metric for significance.

**Response:** Thank you for your comment. While my argument in the Introduction was a comparison of the statistics that can be done over a few decades with only one climate mode, so really no statistics, and the statistics that can be done over many decades and different scenarios

that are provided by ESMs, I now see that the p-value might not be enough. Having taken the time to learn more about the literature in statistics, I have now added the following section to the Methods:

*"2.7 Coefficient of determination, p-values, and Bayes factor*

*To determine the strength of correlations, the coefficient of correlation was calculated throughout this study ($r^2$). In addition, the p-value was calculated to test the hypothesis that the trend change in atmospheric $CO_2$ from one decade to another decade is a significant driver of trends of the global and regional ocean carbon sink. A p-value larger than 0.1 indicates little or no evidence or that hypothesis exists, a p-value from 0.1 to 0.05 indicates weak evidence or a suggestion of evidence, a p-value from 0.05 to 0.01 indicates evidence or modest evidence and a p-value from 0.01 to 0.001 indicates strong evidence (Held and Ott, 2018). In addition, an upper bound for the Bayes factor can be calculated following Halsey (2019). Throughout the manuscript, the p-values are never larger than 1e-89 resulting in Bayes factors that are at least 1e86. Based on the Bayes factor the hypothesis that the trend change in atmospheric $CO_2$ from one decade to another decade is a significant driver of trends of the global and regional ocean carbon sink is 1e86 more likely than the hypothesis that the trend change in atmospheric $CO_2$ from one decade to another decade is not a significant driver of trends of the global and regional ocean carbon sink. As p-values are that small and Bayes factors are that high, I simply refrain to report that the p-values are smaller than 0.001."*

As written in this new section on the manuscript, I have refrained from adding an additional table as the p-values are extremely small and the Bayes factors are extremely large, both indicating strong confidence in the identified relationship. In addition, the relationship is found on explained mechanisms based on McKinley et al. (2020), which further supports the argument that it is not a randomly emerging relationship.

Results:

In Figures 1 and 2 it is hard to tell which line represents which scenario. Please choose colors with more contrast.

**Response:** The colors were chosen in accordance with the IPCC report for the SSPs to be recognizable by most readers. Therefore, I prefer not to change the colors.

In Figures 3-5 it is difficult to tell the difference between the observation-based points and the GOBM-based points. They are both too small (given the black outline), and the colors are too similar to easily differentiate.

**Response:** When I chose the colors orange and pink, I tried to find colors that are easier to distinguish, that are different enough from the viridis colormap used for the ESMs and are also distinguishable for color blind people. During the revisions, I tested different other options but

could not find a better combination of colors. I am happy to change colors if the reviewer or the editor finds a better solution.

In Fig. 3, the dots size was slightly increased. In Figure 5, increasing the dot size is unfortunately not possible as larger dots would hide other dots. The black outline is necessary to guarantee clear visible difference to the dots for the ESMs.

I find Figure 4 very interesting and would have liked more discussion about it. It looks like there may be some regional variation in at what global sink strength the regional sink deviates from the expected trend. Intuitively this makes sense to me, but it would be interesting to see whether it really is the case or just me seeing things. Either way it would add interesting discussion about why the relationship breaks down. Also, considering that the regional analysis, for which no correction of the low bias in sink magnitude was performed (section 2.2), produces results so comparable to the global analysis, why is the bias-correction in section 2.2 necessary? It would be good if it could be tested what the results would be if no correction was done.

**Response:** The test for the difference with and without correction was provided in the answer above. Indeed, the correction has only a small effect. Nevertheless, I still prefer to keep it as it is always better to correct for known biases, even if the effect is rather small. Regionally the $r^2$ are almost all the same as globally (when calculated without correction). The only difference is in the Arctic Ocean, which is already discussed in the manuscript:

*"The correlation coefficient is larger than 0.84 in the Atlantic, Pacific, Indian, and Southern Ocean. Only the Arctic Ocean has a smaller correlation coefficient of 0.66.*

*In the Arctic Ocean, the carbon sink has been shown to be already substantially more affected by climate change than in any other ocean basin (Yasunaka et al., 2023). In the future, when sea ice will disappear and the Arctic will continue to warm faster than any other region, the importance of climate change for the Arctic Ocean carbon sink will likely remain relatively large, for example through freshening (Terhaar et al., 2021a) and a change in the seasonal cycle of $pCO_2$ (Orr et al., 2022), and hence reduce the importance of changes in the atmospheric $CO_2$ for trends in the ocean carbon sink."*

In addition, the three brief episodes of a few years where the relationship does not hold in the Southern Ocean are also mentioned although the overall relationship is strongest in the Southern Ocean despite these three episodes. While these three episodes have no effect on the relationship, an analysis and explanation for this deviation would interesting. In the manuscript, I provide an explanation for two of the three periods, when the deviations are also globally visible, but I cannot provide an explanation for the period from 1995 to 2005:

*"The time periods where the differences are visible globally (2030-2050 and 2080 to 2100 under SSP1-2.6) are the times when the growth in atmospheric $CO_2$ stops and when it starts to decrease in that scenario (Fig 1c). As the atmospheric $CO_2$ growth rate changes quickly in these periods (Fig. 2a), first by changing into a decreasing phase and then transitioning into a stabilizing phase, it appears that a fast transition of the trend change in atmospheric $CO_2$ temporarily leads to differences in the expected relationship. If the trend change in atmospheric*

*CO₂ decreases fast, the trend in ocean carbon sink remains larger than expected and if the trend change in atmospheric CO₂ increases fast, the trend in ocean carbon sink remains smaller than expected. However, the drivers behind the divergence from the expected decadal trend of the multi-model mean in from 1995 to 2005 in the Southern Ocean remain unclear and should be analysed in future research."*

In addition to this assessment, an in-depth analysis of the response in the Southern Ocean for the time period from 1995 to 2005 would be needed, which, however, extends the scope of this manuscript. Hence, no further discussion is added.

Section 4 warrants a more robust statistical analysis and more discussion. Right now no reasons are given for the presented differences. Also, given the short timeframe for most of the observation-based products how robust are the presented results?

**Response:** The difference between the expected trends based on ESMs and $p$CO₂ products will also need much more work, especially on the side of the $p$CO₂ products. This study here focuses on the ESMs and what can be expected in trends. I believe that the ball is now on the side of the $p$CO₂ products to explain why trends are that large in these products. Part of the explanation was given by Hauck et al. (2023) as discussed in the manuscript.

The timeframe should be no problem. As long as these $p$CO₂ products and GOBMs cover at least one decade, that decade can be compared to the expected trend as the driver of that trend, the change in atmospheric CO₂ is known for much longer timeperiods.

Minor comments:

Line 26: add "and" before "the ocean"

**Response:** Changed as suggested.

Line 71: in the abstract you state "3 to 7 decades", here it is "four to seven"

**Response:** Changed to "three to seven".

Line 75: the observations are not just relatively sparse, they are very sparse

**Response:** The word "sparce" was removed following the reviewers' comment.

Line 88-89: this sentence is unclear and needs rewriting for clarity

**Response:** The sentence was modified and divided into two sentences for clarification.

Line 104: move "from phase 6 of the Coupled Model Intercomparison Project (CMIP6)" to directly after "12 ESMs" on line 103

**Response:** Changed as suggested.

Line 158: change "effect" to "affect"

**Response:** Changed as suggested.

Line 190: It is not easy to see these jumps in the figure. Consider highlighting them somehow (different colors?)

**Response:** The jumps are now marked in the figure as suggested. Please see response to major comment of reviewer 2 (Prof. Galen McKinley).

Line 240: replace the first "and" by "with", and the second "and" by "or"

**Response:** Changed as suggested.

Line 261-262: This sentence is incomplete

**Response:** The sentence was changed to:

*"Once the atmospheric $CO_2$ growth declines, the trend in the ocean carbon sink becomes negative."*

Line 305: It is unclear whether this is the Pearson correlation coefficient (r) or the coefficient of determination (r^2). Based on the rest of the section I would guess the latter, but please specify and use the correct terminology.

**Response:** Changed as suggested.

Line 322-323: This sentence (beginning with "As the atmospheric …") is incomplete

**Response:** The sentences was corrected to:

"As the atmospheric $CO_2$ growth rate changes quickly in these periods (Fig. 2a), first by changing into a decreasing phase and then transitioning into a stabilizing phase, it appears that a fast transition of the trend change in atmospheric $CO_2$ temporarily leads to differences in the expected relationship."

Line 351: earlier in the manuscript "we" is used – be consistent

**Response:** "We" was changed to "I" as suggested.

Line 378: add "final" before GOBM

**Response:** Changed as suggested.

Line 390: "causes changes"

**Response:** Changed as suggested.

Figure 6: Add details in caption about the vertical line and shading in subplot c)

**Response:** The details were added as suggested.

Figure 9: It is next to impossible to tell the lines on this figure apart. Please choose different colors. I would also recommend making the lines for individual models thinner.

**Response:** The thin lines were intended to show the range of the individual models. As the previous version of the figure has not succeeded in doing so, the thin lines were replaced by a lighter shading that indicates the maximum and minimum of the variability of the decadal trends in the ESMs. The revised figure looks as follows:

[Figure]

**Figure 9: Variability of the decadal trends of the zonally integrated ocean carbon sink in earth system models.** The multi-model mean (thick blue line) and the 1-σ standard deviation of the variability of the zonally integrated ocean carbon sink across the 251 years of the pre-industrial control simulation across all 12 ESMs. In addition, the maximum and minimum variability in the ESMs are shown at each latitude (thin blue lines).

Line 563: Are these the same five in every decade? Please specify and if not this warrants more discussion

**Response:** Following the suggestion of the reviewer, the following sentences were added to the manuscript:

[revised manuscript text omitted]

---

## Author Comment (AC2)

I am deeply thankful to Prof. Galen McKinley for her constructive and very helpful evaluation of the manuscript, which has strongly improved its quality. Below, I have addressed the points that she has raised point by point. Prof. McKinley's text is shown in black and my responses in blue. Text that will be part of the revised manuscript is shown in italic and blue color.

Dr. Terhaar studies CMIP6 historical and future projections to assess the drivers of trends in the global ocean carbon sink. The author compares decadal trends in atmospheric pCO2 to decadal trends in the ocean sink, and proposes a 1 decade lag of the ocean behind the atmosphere. A mechanistic explanation for this lag is not offered. The author also proposes that this analysis of ESMs demonstrates that the decadal trends in pCO2 products are too large.

Major comments

- The a 1 decade delayed response of the ocean sink to trends in the atmospheric growth rate in CMIP6 is intriguing. But why? A proposed mechanism for this effect is missing in the manuscript. In McKinley et al. 2020, we show that change in the atmospheric growth rate impacts the ocean sink with no lag, due to the impact on the delta pCO2. Lovenduski et al. (2021) demonstrate this mechanism with CANESM5 for changes in the atmospheric growth rate consistent with COVID19. What is different mechanistically, or about the analysis performed here, that leads to a very different conclusion here? The author needs to address this directly.

**Response:** I believe this is a misunderstanding. What I proposed here is not that the decadal trend in the ocean carbon sink is driven by the trend in the atmospheric $CO_2$ (= atmospheric $CO_2$ growth rate) in the preceding decade, i.e., a 1-decade delayed response. Instead, I propose that it is the trend in atmospheric $CO_2$ in this decade compared to the trend in atmospheric $CO_2$ in the previous decade that drives the trend in the ocean carbon sink. As such, it is not a 1-decade delayed response but includes information from both decades.

The here proposed mechanism is in line with Lovenduski et al. (2021) and McKinley et al. (2020). McKinley et al. (2020) write in their abstract: "First, the global-scale reduction in the decadal-average ocean carbon sink in the 1990s is attributable to the slowed growth rate of atmospheric $pCO_2$. The acceleration of atmospheric $pCO_2$ growth after 2001 drove recovery of the sink." In their study, they define a slowed down trend with respect to a prescribed linear trend. However, computing the difference to a linear trend works not for the entire historical period where $pCO_2$ growth is exponential so that a linear trend, even over 30 years, would always lead to a too small $pCO_2$ growth first and a faster $pCO_2$ growth faster, and would also not work when atmospheric $pCO_2$ growth peaks and changes from an increase to a decline.

Here, I use the mechanism of the slowing and acceleration of define the slowing or acceleration of atmospheric $pCO_2$ growth as the driver of changes in the ocean sink on decadal trends but define slowing and acceleration not with respect to a theoretical linear trend but compared the atmospheric $pCO_2$ growth in the preceding decade. Changes in decadal trends, as opposed to shorter trends, are used because inter-annual variability in the ocean carbon sink and the atmospheric $pCO_2$ growth disguise the trends on shorter timescales as shown by Lovenduski et al. (2021). Throughout the manuscript, I show that this definition of slowing and acceleration of

the atmospheric $p$CO$_2$ growth with respect to the preceding decade works well for low- to medium CO$_2$ emission scenarios.

In the revised manuscript, I will clarify this as follows to avoid any further misunderstanding:

*"Although neither the atmospheric CO$_2$ nor its growth rate can quantify the strength of the ocean carbon sink various time period and different trajectories of atmospheric CO$_2$, the atmospheric CO2 growth rate can nevertheless be used to understand changes in the ocean carbon sink on decadal timescales, i.e., decadal trends of the ocean carbon sink. For the period from 1980 to 2018, it has been shown that a slowing of the growth rate in comparison to a linear trend has led to a stagnation of the increase of the ocean carbon sink and that an accelerated increase of the growth rate has led to a strongly increasing carbon sink (McKinley et al., 2020)."*

*[...]*

*As a slowing or acceleration of the growth rate in comparison to a theoretical linear trend as in McKinley et al. (2020) is not anymore possible over longer time periods of exponential growth or when atmospheric CO$_2$ peaks, I here generalize the idea of McKinley et al. (2020) that a slowing or acceleration of the atmospheric CO$_2$ growth rate drives the trends of the ocean carbon sink by defining such slowing or acceleration as the difference in the growth rate in a given decade with respect to the preceding decade. When defining slowing or acceleration of the atmospheric CO$_2$ growth rate that way, a clear relationship ($r^2$=0.91) emerges indeed over the entire historical period and all four future scenarios over the 21$^{st}$ century (excluding years where the ocean carbon sink exceeds 4.5 Pg C yr$^{-1}$) between changes in the atmospheric CO$_2$ growth rate and the decadal trend of the multi-model average of the ocean carbon sink (Fig. 3)."*

- What is the impact on the findings of the adjustments to model output following Terhaar et al. (2022)? Dr. Terhaar and colleagues' previous findings are interesting, but not conclusive. Others, such as Goris et al. (2018) propose alternative metrics for such a constraint. It is important to understand the impact of this adjustment on these results.

**Response:** The alternative metrics presented by Goris et al. (2018) are not opposite to those found by Terhaar et al. (2022b). Instead, they are complimentary and indicate similar biases in the models, as also shown for ocean-biogeochemical models in hindcast mode by Terhaar, Goris, et al. (2024). Below, I have re-made the main figure from the manuscript without the adjustment. Without the bias-adjustment for each Earth System Model, the strength of the relationship slightly reduces from $r^2$=0.90 to $r^2$=0.83 but remains strong and significant. Hence, the findings are not sensitive to the bias-adjustment but the bias-adjustment still yields more reliable results as the studies from Goris et al. (2018), Terhaar et al. (2022b) and Terhaar, Goris, et al. (2024) suggest. For clarification, I added the following sentence to the Methods:

*"The adjustment corrects for known biases in the models' circulations and surface ocean carbonate chemistry and hence reduced differences in the overall magnitude of the simulated carbon sink between ESMs (Terhaar et al., 2022b). This reduction in the difference in the magnitude of the carbon sink also reduces differences between the magnitude of trends and*

*slightly improves the relationships found here as it ($r^2$ in Figure 3 would have been 0.83 without adjustment instead of 0.91 with adjustment). Nevertheless, the results would quantitatively and qualitatively almost identical with and without that adjustment. "*

**(a)**                                              **(b)**

[Figure]

**Figure 3:** The relationship between changes in the atmospheric CO2 growth rate and decadal trends of the global ocean carbon sink for the multi-model mean. **(a)** Decadal trends of the multi-model mean ocean carbon sink compared to changes in decadal trends in atmospheric CO2, which represent the decadal averaged growth rate of atmospheric CO2. The dark blue to yellow circles without a surrounding black line show multi-model averages for all years of the historical period from 1850 to 2014 and for all years from 2015 to 2100 for all four SSPs. All decades over from 1850 to 2100 are shown, i.e., 2000-2009, 2001-2010, 2002-2011, etc. The brown line shows a linear fit for all years when the global ocean carbon sink is smaller than 4.5 Pg C yr-1 and the brown shading is the 1-s projection uncertainty. The dots with black lines around them show values from the respective ensemble means of the pCO2 products (pink) and GOBMs (orange) from the Global Carbon Budget 2023 (Friedlingstein et al., 2023) for the three decades between 1990 and 2020. **(b)** The simulated ocean carbon sink in comparison to the expected ocean carbon sink based on the relationship in **(a)** and the prescribed trend change in atmospheric CO2 in the simulations.

- The author focuses the introduction on the weaknesses of pCO2 products and GOBMs, but does not adequately acknowledge that ESMs also have weaknesses. The fact that the author will adjust and detrend the ESMs before doing his analysis needs to be acknowledged here, as just one example of a weakness. Please adjust this discussion to be more balanced.

**Response:** As suggested by the reviewer, we have added the following text about ESMs to the Introduction:

*"Here, I use an ensemble of 12 ESMs to provide a new perspective on potential drivers of the decadal trends of the ocean carbon sink from phase 6 of the Coupled Model Intercomparison Project (CMIP6) (Table 1). Fully coupled ESMs are another tool to quantify and understand the ocean carbon sinks (e.g., Joos et al., 1999; McNeil and Matear, 2013; Frölicher et al., 2015; Goris et al., 2018; Terhaar et al., 2022b, 2021b). As ESMs are fully coupled and not forced with atmospheric reanalysis data, they do not simulate the same inter-annual internal climate variability as pCO2 products and GOBMs do and their biases of the surface ocean physics and biogeochemistry are thus larger than surface ocean biases of GOBMs (Terhaar et al., 2022b;*

*Terhaar et al., 2024). However, ESMs have distinctive advantages compared to pCO2 products and GOBMs for the analyses of decadal drivers of the ocean carbon sink because (1) they cover a period of 251 years from 1850 to 2100, (2) cover at least four different future scenarios, and (3) they all have a different internal climate state.”*

- The author needs to be more precise about ESMs vs. GOBMs. Papers such as Gruber et al 2023 and the associated literature, as well as RECCAP2, focus on comparing pCO2 products to GOBMs, not to ESMs as indicated on Line 458 and below. Please check throughout the paper and make sure the discussion does not confuse.

**Response:** I believe there has been a misunderstanding. Although Gruber et al. (2023) did not focus on ESMs, but on pCO2 products and GOBMs as explained by the reviewer, their discussion extends to ESMs (here called coupled carbon-climate models):

"An ocean sink that is more sensitive to climate change than currently assumed in coupled carbon-climate models[52] would imply that the ocean will take up less $CO_2$ from the atmosphere in the future than anticipated."

Reference 52 (Arora et al., 2020) in Gruber et al. (2023) is about idealized scenarios with steadily increasing atmospheric $CO_2$ and carbon-climate feedbacks. However, here I have shown that the variability of the ocean carbon sink is mainly driven by variability in atmospheric $CO_2$ growth, which does not exist in these idealized scenarios, and not to climate change. Furthermore, I have demonstrated in Fig. 7 that the ESMs simulate equal or larger trends in the major climate modes. Moreover, several past studies have shown that the variability and trends in $pCO_2$ products are instead overestimated (e.g., Gloege et al., 2021; Hauck et al., 2023) further questioning the discussion above by Gruber et al. (2023).

Hence, I believe that the hypothesis by Gruber et al. (2023) that the decadal trends in the ocean carbon sink in ESMs is too small and that this means that the sensitivity to climate change is to small is not supported by Arora et al. (2020) and that the information in this manuscript challenges that hypothesis. Hence, I think that sentence in line 458 was correct (*"Thus, there is no indication that the decadal variability of the ocean carbon sink in ESMs (Fig 7a) might be too small because of a too small internal climate variability in ESMs as previously hypothesized (Gruber et al., 2023)”*).

To avoid any further misunderstandings, I have nevertheless changed it to:

*"As the decadal trends in climate mode are larger or equal to the observed ones, there is no indication that the decadal variability of the ocean carbon sink in ESMs (Fig 7a) might be too small because of a too small internal climate variability in ESMs as previously hypothesized by Gruber et al., (2023) based on small carbon-climate feedbacks in idealized scenarios without variability in the atmospheric $CO_2$ growth (Arora et al., 2020).”*).

In addition, I have read the manuscript carefully again to make sure that all statements are correct.

- There is a lot of discussion of detailed features from the figures that are very difficult for the reader to see due to a lack of annotation. For example, on figures 1 and 2 where atmospheric CO2 concentration or growth rate are plotted against the ocean sink, the author discusses features at specific dates. It is not possible to see these dates on such a figure. The author needs to make sure the reader can identify the features he discusses.

**Response:** As suggested by the reviewer, the discussed time periods are now marked on the respective panels as suggested by the reviewer. The figures were revised to:

[Figure]

**Figure 1: The relationship between atmospheric CO₂ and the global ocean carbon sink. (a)** The annually averaged atmospheric CO₂ that was used to force the ESMs from CMIP6 based on observation-based estimates from 1850 to 2014 (black) and based on four different SSPs (SSP1-2.6 in blue, SSP2-4.5 in orange, SSP3-7.0 in red, and SSP5-8.5 in brown) from 2015 to 2100. **(b)** The resulting ocean carbon sink as simulated by 12 ESMs (Table 1) after being adjusted for biases in circulation and surface ocean carbonate chemistry following Terhaar et al. (2022). The thick lines indicate multi-model means and the shading the 1-σ standard

deviation across the model ensemble. Relationships between atmospheric $CO_2$ and the annually averaged ocean carbon sink **(c)** for the historical period until 2014 and **(e)** for the 21$^{st}$ century from 2015 onwards, as well as between atmospheric $CO_2$ and the cumulative ocean carbon sink **(d)** for the historical period until 2014 and **(f)** for the 21$^{st}$ century from 2015 onwards. The light grey shadings in **(a)** – **(d)** indicate the time periods from 1920 to 1960 and from 1990 to 1995.

[Figure]

**Figure 2: The relationship between the atmospheric $CO_2$ growth rate and the global ocean carbon sink. (a)** The annually averaged atmospheric $CO_2$ growth rate based on atmospheric $CO_2$ forcing files from CMIP6, which are based on observation-based estimates from 1850 to 2014 (black) and based on four different SSPs (SSP1-2.6 in blue, SSP2-4.5 in orange, SSP3-7.0 in red, and SSP5-8.5 in brown) from 2015 to 2100. **(b)** The ocean carbon sink as simulated by 12 ESMs (Table 1) after being adjusted for biases in circulation and surface ocean carbonate chemistry following Terhaar et al. (2022). The thick lines indicate multi-model means and the shading the 1-σ standard deviation across the model ensemble. Relationships between atmospheric $CO_2$ growth rate and the annually averaged ocean carbon sink **(c)** for the entire period from 1850 to 2100 and **(d)** only for historical period until 2014. The light grey shading in **(a)** indicates the period where direction atmospheric $CO2$ observations are available and the pale green shading in **(a)** and **(b)** and the pale green dots in **(c)** and **(d)** indicate the 1990s and 1940s. The zero growth rate and ocean carbon sink in **(a)** and **(b)** are shown as black dashed lines.

Minor

Line 33 - 12 members is not a "large ensemble" in the common understanding of this terms. This would be at least many 10s of members. Please revise.

**Response:** The sentence was changed to "The robust relationship over an ensemble of 12 different ESMs" as suggested by the reviewer.

Line 59 – Ridge and McKinley 2021 Biogeosciences should also be cited

**Response:** The reference was added as suggested.

Line 78 – Please add Gloege et al. 2022 JAMES, Bennington et al. 2022 GRL, Bennington et al. 2022 JAMES

**Response:** The references were added as suggested.

Line 80 – Please add LaCroix et al. 2020

**Response:** The reference was added as suggested.

Line 87 – Says "pCO2 products" here, should be GOBMs

**Response:** Thank you. The mistake was corrected as suggested.

Line 93 – unclear to what "both products" refers

**Response:** For clarification, "Both products" was replaced by "$pCO_2$ products and GOBMs".

Line 94 – Gloege et al. 2021 did not use GOBMs; large ensembles of ESMs were used.

**Response:** Thank you. The mistake was corrected as suggested.

Line 110 – Please reference McKinley et al. 2023 ERL as a study that considers the full CMIP6 suite.

**Response:** The reference was added as suggested.

Line 193 – Clarify here briefly that 2014 is the end of the historical period of forcing - i.e. "After 2014, when SSP scenario forcing begins,… " or similar

**Response:** As suggested by the reviewer, the beginning of the sentence was changed to:

*"After 2014, when the historical period in CMIP6 ends and SSPs start,…"*

Line 218 – A square root relationship should be dependent on the units. Please include units. Please also mark this square root relationship on the figure

**Response:** As the sentence appeared to have created more confusion than clarity and the relationship was weak, we have removed any mention such a square root relationship from the manuscript.

Line 225 – "ocean carbon sink does not go back close to zero but remains almost stable (Fig. 2b)." This cannot be easily seen on the plot

**Response:** For clarification, dashed lines were added where the atmospheric $CO_2$ growth rate (Fig. 2a) and ocean carbon sink (Fig. 2b) are zero (please see revised Fig. 2 above).

Section 2.1 Some of these ESMs provide multiple ensemble members to CMIP6. How are the ensembles used here? Just the first one taken? An average made? If the latter, then it could impact results by averaging out some of the internal variability of individual members, and this would need to be discussed. Please make this clear, and discuss any impacts on results.

**Response:** Only the first ensemble member is used. The reasoning is now discussed in the revised manuscript:

*"For each ESM, only the first ensemble member is used as averaging over multiple ensemble members would have removed variability and using different numbers of ensemble members per ESM would have biased results towards the ESMs with more ensemble members."*

Section 3.2, and Line 251-262. It is difficult to follow these discussions. Adding annotation on figures (as suggested below), increasing the size of the figures so that these features can be seen, and/or revising the text to more clearly describe the features being discussed.

**Response:** The discussed time periods are now marked on the respective panels as suggested by the reviewer. Please see response to major comment above.

Line 245-246. The first phrase of this sentence is incomplete, and also and above it was said there is a square root relationship. Please revise.

**Response:** The sentence was revised as suggested by the reviewer, and any mention of a square root relationship was removed.

Line 260. Replace "done" with "down"

**Response:** The word was changed as suggested by the reviewer.

Line 264. It is not true that it is "not possible" to compare to a linear trend. For example, it would be possible to calculate linear trends for 30 years, and use this as comparison. The author needs to find a better way to justify the approach taken here.

**Response:** I disagree with the reviewer. The linear trend might be possible as long as the increase in atmospheric $CO_2$ can be approximated by a linear trend. Where this is not possible, i.e., when atmospheric $CO_2$ peaks or the increase is strongly exponential, this is not possible. The sentence has been revised following a major comment above by a reviewer.

Line 270. What is the mechanism of this delayed response?

**Response:** As described in the first major comment, I believe that is a misunderstanding that has been clarified in the response to that comment.

Line 318. SSP1-2.6 scenario is not clearly labeled in figure 3a. Please ensure the reader can follow this discussion.

**Response:** The periods are now marked in the figure. The revised figure looks as follows:

[Figure]

**Figure 3: The relationship between changes in the atmospheric CO₂ growth rate and decadal trends of the global ocean carbon sink for the multi-model mean. (a)** Decadal trends of the multi-model mean ocean carbon sink compared to changes in decadal trends in atmospheric CO₂, which represent the decadal averaged growth rate of atmospheric CO₂. The dark blue to yellow circles without a surrounding black line show multi-model averages for all years of the historical period from 1850 to 2014 and for all years from 2015 to 2100 for all four SSPs. All decades over from 1850 to 2100 are shown, i.e., 2000-2009, 2001-2010, 2002-2011, etc. The brown line shows a linear fit for all years when the global ocean carbon sink is smaller than 4.5 Pg C yr⁻¹ and the brown shading is the 1-σ projection uncertainty. The dots with black lines around them show values from the respective ensemble means of the pCO₂ products (pink) and GOBMs (orange) from the Global Carbon Budget 2023 (Friedlingstein et al., 2023) for the three decades between 1990 and 2020. Small deviations from the relationship in SSP1-2.6 are marked by 'SSP1-2.6'. **(b)** The simulated ocean carbon sink in comparison to the expected ocean carbon sink based on the relationship in **(a)** and the prescribed trend change in atmospheric CO₂ in the simulations.

Line 390. Strike "to"

**Response:** The word "to" was removed as suggested by the reviewer.

Line 455-482. ESMs may get the climate modes, but not necessarily the ocean carbon sink response to these modes. The following discussion of biogeochemical vs physical driven variability in GOBMs and of higher resolution models does try to address this, but still it is not conclusive because these are still models being used as the point of comparison. There is circularity in this discussion that needs to be acknowledged – i.e. though sampling of pCO2 may lead the pCO2 products to overestimate decadal variability, it also remains possible that the pCO2 products are capturing real signals that we are not modeling.

**Response:** In the revised manuscript, it will be acknowledged that it remains possible that the *pCO₂ products are capturing real signals that we are not modeling as suggested by the reviewer:*

*"Overall, the dominance of physical variability over biogeochemical variability and the larger decadal trends of climate modes in ESMs than in the real world suggest that the ESMs do not*

*underestimate the natural variability of the ocean carbon sink although it always remains possible that the pCO2 products are capturing real signals that are not yet simulated."*

Line 486. "ESMs used here"

**Response:** The words were reordered as suggested by the reviewer.

Line 523. Again, what is the mechanism of the decadal lag in the ocean response to the atmospheric growth rate?

**Response:** As described in the first major comment, I believe that is a misunderstanding that has been clarified in the response to that comment. For further clarification, this sentence was changed to:

*"While McKinley et al. (2020) have focused on the last decades and suggested that the trends in the ocean carbon sink depends on differences on the atmospheric growth rate of $CO_2$ compared to the long-term trend of the growth rate, I could generalize this idea here and show that it is the change in the growth rate compared to the previous decade that drives the trends of the ocean carbon sink over a wide range of timescales and SSPs."*

Figure 1.

- Please add a legend on the figure. Please make the blue more clearly distinguishable as not black/gray.
- Consider marking 1920, 1960, 1990, 2000 on panel c, d. This is needed to more easily follow the discussion about "jumps" at line 189-191.

**Response:** The periods are now marked as suggested by the reviewer (please see Figure above). However, the colors were not changed as these are the official colors from the IPCC reports that I prefer to keep making the SSPs easier recognizable.

Figure 2

- It is not clear where 1920 and 1940 are in panels c and d; please these clearly mark on the plots

**Response:** The annotations were added as suggested.

Figure 3

- The text suggests that here it is ocean sink in decade 2 compared to atm CO2 trend in decade 1, but it isn't stated in this way in the caption. The caption suggests they are concurrent trends. This needs clarification.

**Response:** I believe this misunderstanding has been clarified in the response to the major comments.

**References**

[revised manuscript text omitted]

---

## Author Response (AR2)

I am very thankful to Dr. Siv Lauvset for her two constructive and very helpful evaluations of the manuscript, which have strongly improved its quality. Below, I have addressed the points that were raised by Dr. Lauvset in the second review round point by point. The reviewer's text is shown in black and my responses in blue.

**Reviewer 1: Dr. Siv Lauvset**

The author has revised the manuscript according to my first review, and the revised version reads very well and is ready for publication. I still think some of the figures are hard to interpret due to the choice of colors, but I respect the author's desire to align with the IPCC color scheme. I only have a few technical comments (detailed below).

**Response:** Thank you for your positive and constructive evaluation of the manuscript.

Line 115: It is unclear whether the "robustness of drivers" refers to the robustness of the drivers themselves, or the robustness of determining what the drivers are. Please clarify.

**Response:** For clarification the sentence was changed in the revised manuscript as follows:

*"The long time-period with different climate states in each model gives ample material to perform statistical analyses and the different future scenarios allow to test how robustly potential drivers predict the decadal variability of the ocean carbon sink under continuously rising and under strongly decreasing carbon emission trajectories."*

Line 125: Replace "thus" with "this"

**Response:** Changed as suggested.

Line 153-155: The sentence is incomplete.

**Response:** The two last words of the sentence were removed to correct the sentence. It now reads:

*"This reduction in the difference in the magnitude of the carbon sink also reduces differences between the magnitude of trends and slightly improves the relationships found here."*

Line 155: replace "would" with "are"

**Response:** Changed as suggested.

Figure 2: In the caption it says that green shading indicate the 1990s and 1940s. The correct is that they indicate the 1920s and 1940s.

**Response:** Changed as suggested.

Line 335: move "large" after "five"

**Response:** Changed as suggested.

Line 349: Misspelling - should be SSP1-2.6

**Response:** Changed as suggested.